# Optically driven control of mechanochemistry and fusion dynamics of biomolecular condensates via thymine dimerization

Vahid Sheikhhassani[1,2], Faith H. K. Wong [1,2,3], Daniel Bonn [4], Jeremy D. Schmit[5] & Alireza Mashaghi [1,2] ✉

Phase-separated biomolecular condensates are functional elements in cells, contribute to protocell formation in prebiotic systems, and represent a distinct class of soft matter. Controlling condensate mechanochemistry is critical for function and material properties. Although photochemical processes are widespread in nature and can be harnessed in engineering, it remains unclear how condensate formation affects photochemistry, and conversely how photochemistry alters condensate dynamics. Using scanning probe microscopy combined with UV-controlled photochemistry and optical imaging, we develop assays to probe mechanical transitions and fusion dynamics in condensate droplets, revealing that UV-induced thymine dimerization alters condensate nucleation and coalescence. Depending on the frequency and topological arrangement of thymine dimers, particularly the balance between inter- and intrachain crosslinks, UV can drive transitions from liquid-like to solid-like states or induce aggregates. UV also promotes arrested fusion and stable compartmentalization of droplets, resilient to environmental changes and with implications for prebiotic chemistry and bio-inspired engineering.

Biomolecular condensates formed through liquid-liquid phase separation (LLPS) play key roles in various cellular processes, ranging from gene regulation to stress granule formation[1–5]. These condensates assemble through multivalent interactions among macromolecules such as nucleic acids and proteins[6–9], and while physiological condensates are often liquid-like, they can exhibit diverse material and mechanical properties, particularly upon aging or under stress conditions that alter their interactions[10–15]. The strength, density, and topological arrangement of interactions, along with the intrinsic mechanical properties of the polymer chains, are critical determinants of condensation and the resulting mechanical phenotype[16]. Such variations in condensate material state may directly influence molecular transport and chemical reactions within their microenvironments, effects that are central to cellular biochemistry and exploitable for reaction control in engineered systems[6,17–21]. Modulation of the mechanical properties of condensates is typically achieved by introducing new molecules (e.g., crosslinkers) or adjusting the concentrations of molecules and ions, approaches that inherently

[1]Medical Systems Biophysics and Bioengineering, Division of Systems Pharmacology and Pharmacy, Leiden Academic Centre for Drug Research, Leiden University, Leiden, The Netherlands. [2]Laboratory for Interdisciplinary Medical Innovations, Centre for Interdisciplinary Genome Research, Leiden University, Leiden, The Netherlands. [3]Department of Life Sciences, Faculty of Natural Sciences, Imperial College London, London, UK. [4]Van der Waals-Zeeman Institute, Institute of Physics, University of Amsterdam, Amsterdam, The Netherlands. [5]Department of Physics, Kansas State University, Manhattan, KS, USA. ✉e-mail: a.mashaghi.tabari@lacdr.leidenuniv.nl

alter condensate composition, making it difficult to disentangle the effects of mechanics and chemistry[22].

Light can be exploited to modulate material properties in soft and polymeric systems[23,24]. Yet, light-triggered control over the mechanical state of condensates, achieved without significantly altering their chemical composition, offers a largely unexplored route for engineering condensates. One particularly relevant form of light stimulus is ultraviolet (UV) radiation, which is both a major cause of DNA damage through the formation of covalently bonded pyrimidine dimers such as thymine dimers (TT) and a critical factor in the emergence of life on Earth[25,26]. TT dimers disrupt normal DNA replication[27] and transcription[28], leading to mutations that can initiate various diseases, and are the primary cause of skin cancer[29]. Cells recognize and attempt to repair these lesions[30]; however, persistent or improperly repaired TT dimers can give rise to short stretches of single-stranded DNA (ssDNA), ultimately contributing to genomic instability and carcinogenesis[31]. Furthermore, UV radiation was abundant on early Earth and contributed to the synthesis of building blocks of life[32], while coacervate condensates of these building blocks are also considered plausible protocellular structures[33]. The droplets feature a highly dynamic molecular environment[34], and are permeable to molecules from their surroundings with certain degrees of selectivity, enabling chemical synthesis and molecular recognition (e.g., receptor-ligand interactions) required for life[35]. Nucleic acids, known for their propensity to form condensates, were likely key components of protocells[33]. Because coacervates lack lipid membranes, these protocellular structures provide only limited shielding of their contents from solar UV. It remains unclear, however, what UV does to (nucleic acid) coacervates and whether it can shape their structural mechanics.

A technical challenge in studying the mechanochemistry of condensate droplets is the difficulty in probing viscoelastic and fusion properties as they undergo liquid-to-solid transitions. Active microrheological characterization of condensate droplets is mostly performed utilizing optical tweezers (OT)[36,37], which are constrained to low-force regimes and generally rely on trapped beads. Micropipette aspiration is also constrained by achievable force (suction pressure) and experimental throughput[38,39]. Recently, scanning probe microscopy (SPM), a technique widely used in mechanobiology and materials science[40], has been applied to characterize the material properties of condensates[22,41,42], covering a significantly higher range of forces and eliminating the need for exogenous particles in the system. Yet, SPM has never been applied to study droplet-droplet fusion, a gap that, if addressed, could enable direct insights into condensate mechanics and interactions.

Here, we introduce a photochemical strategy to precisely modulate condensate mechanics using UV-induced thymine dimerization, enabling control over mechanical properties while preserving chemical composition, that is, without adding new molecules and without changing atomic composition. Using poly-thymine single-stranded DNA (dT40, Table S1) and poly-L-lysine (PLL) as a model system, we demonstrate how this strategy drives transitions among distinct mechanical states, from liquid-like droplets to more rigid, gel- or solid-like networks. To capture these mechanical changes, we developed SPM-based assays combined with optical imaging for label-free characterization of condensate mechanics and fusion dynamics. These transitions lead to dramatic changes in fusion behavior, ultimately driving compartmentalization, a process relevant to prebiotic chemistry and bio-inspired materials design. We attribute these observations to the formation and topological arrangement of UV-induced TT dimers, which can be local, long-range, or occur between chains. Our study provides a compelling example in which a photochemical reaction alters the condensed phase, and conversely, condensation modulates photochemical reaction outcomes.

## Results

### UV irradiation affects the phase separation behavior and promotes arrested fusion droplets

We first explored the phase diagram of PLL/dT40 by varying component concentrations and selected a 1:1 PLL:dT40 ratio at 30 μM of each component for subsequent analyses (Figs. 1A and S1), as this condition produced micron-sized droplets that were well suited for mechanical measurements and imaging. Prior to investigating the effects of UV irradiation, we characterized the baseline phase separation behavior of PLL/dT40 under varying salt conditions, from 0 M to 1 M. No droplets formed at either extreme: high salt screens electrostatic interactions below the threshold for phase separation, while low salt favors tight 1:1 heterodimers with a strong positive net charge (due to charge asymmetry −40 vs. +150), preventing further association. Intermediate salt concentrations, by contrast, allow molecules to exchange partners within a liquid environment[43,44], promoting spherical condensates of varying size. The largest droplets were consistently observed at 150 mM KCl (Fig. 1B, C), a condition used for the remainder of our study.

For UV treatment, samples were prepared over a total period of 4 hours and placed under a UVC lamp (UVP UVG-11, λ=254 nm; 1120 μW/cm² at the sample) (Fig. 1D), with UV illumination applied during the final 30 (+UV30) or 45 (+UV45) minutes[45]. Spectroscopic analysis indicates that UV induces TT dimerization in dT40 under this exposure condition (Fig. 1G, H). Although the UV illumination times used in this study are much longer than the picosecond timescale required for the formation of intramolecular TT dimers[46], it is comparable to the timescale characterizing intermolecular crosslinking[45]. Phase-contrast microscopy immediately after irradiation revealed altered droplet morphologies with an apparent reduction in circularity (Fig. 1E, F). Similar UV-induced morphological changes were observed with droplets formed on nonsticky surfaces (Fig. S2A and S2B), suggesting that observed patterns are primarily driven by UV-induced chemical changes rather than surface adhesion effects (Fig. S2C). To confirm that UV irradiation indeed triggered TT dimer formation inside droplets, we performed immunostaining using anti-TT dimer antibody, which verified TT dimerization in dT40 under these exposure conditions (Fig. 1I).

Our quantitative analysis revealed that extended UV exposure (from 30 to 45 min) led to a higher occurrence of doublets and clustered configurations (Fig. 2A, B), as well as a significant increase in droplet-droplet contact area, measured by the chord length (L) (Fig. 2A, C). This reflects slower relaxation due to the expected increase in viscoelasticity and allows doublets with larger chord lengths to be more readily captured during imaging. Additionally, our particle analysis revealed a significantly lower circularity index of +UV45 droplets compared to +UV30 and control (Fig. 2D). Size distribution analysis of singlet droplets revealed that control samples followed an exponential distribution (rather than log-normal[47] or power law[48] behaviors), consistent with a quench-then-coalesce mechanism[49,50]. However, +UV30 samples, and more prominently +UV45 samples, exhibited a clear shift to a power-law distribution, suggesting a transition to an addition of mass growth model[48] (Figs. 2E and S3).

### Rheological analysis reveals liquid-to-solid-like transitions upon UV exposure

UV exposure can induce inter- and intra-chain thymine dimers, as confirmed by our spectroscopic and imaging analyses, potentially altering condensate mechanics[51,52]. To investigate this, we performed oscillatory rheology measurements of the storage modulus (G′) and loss modulus (G″) on control and UV-illuminated samples using a recently developed SPM-based approach[22] (Fig. 3A and Note S1). Across the entire frequency range examined (1 Hz–100 Hz), G″ consistently exceeded G′, indicating that the control droplets exhibited predominantly viscous (liquid-like) characteristics (Fig. 3B). We note

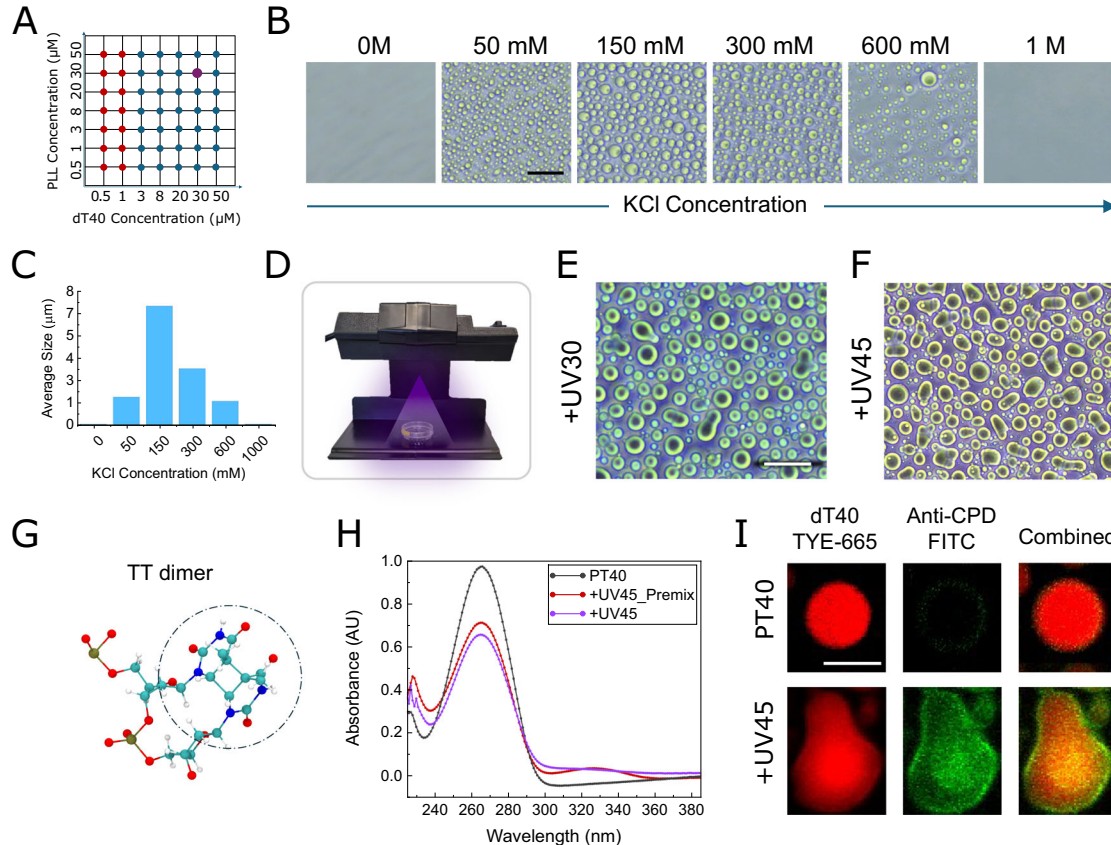

Fig. 1 | **Morphological and chemical characterization of PLL:dT40 condensates with and without UV illumination. A** Phase diagram illustrating condensate formation at different PLL and dT40 ratios. Blue dots represent conditions where droplet formation was observed, while red dots indicate no droplet formation. A selected condition (30 µM PLL: 30 µM dT40) highlighted in purple was used in all subsequent experiments. **B** Effect of salt concentration on condensate formation. No condensates were observed at extreme salt concentrations (0 M and 1 M KCl), while intermediate KCl concentrations resulted in the formation of droplets. **C** Quantification of average droplet diameters at different salt concentrations. The highest average droplet diameter was recorded at 150 mM KCl. **D** UV irradiation setup. A UVC lamp installed on a custom-built stand ensured uniform exposure. All samples were placed in a fixed position on the holder beneath the lamp. Images were captured after 4 h incubation following mixing, including the final 30 or 45 min of UV irradiation (+UV30, +UV45). **E, F** Phase-contrast microscopy showed clear morphological changes in the droplets after 30 and 45 min of UV exposure, with elongated droplets appearing after UV exposure. These elongations were more pronounced following 45 min of irradiation. **G** Ball-and-stick representation of an example TT dimer structure (depicted from PDB structure 9j8w). **H** UV-vis absorption spectra of dT40 before and after UV exposure. The non-irradiated dT40 (black line) displays the characteristic absorption maximum near 260 nm,

corresponding to the π-to-π* transitions of the thymine bases in ssDNA. Following UV exposure, both spectra exhibit a hypochromic effect, a pronounced decrease in absorbance at 260 nm, indicating the formation of thymine-thymine (TT) cyclobutane dimers and the associated disruption of the electronic environment of the bases. Furthermore, a weak shoulder appears between 300 nm and 320 nm in the irradiated spectra, consistent with the formation of minor photoproducts, such as the (6–4) photoproducts or their Dewar isomers. Critically, the reduction in 260 nm absorbance and the intensity of the 300–320 nm shoulder are more pronounced in the +UV45 sample compared to the +UV45_Premix sample (irradiated before mixing with PLL). **I** Immunofluorescence reveals the formation of TT dimers within UV-irradiated condensate droplets. To directly probe this chemical change, condensates were stained with an anti-TT dimer (Anti-CPD) antibody (FITC-labeled). The PT40 samples (top row) confirm ssDNA assembly (red) but show no CPD signal (green). In contrast, the +UV45 sample (bottom row) displays a strong CPD signal co-localized with the dT40 (yellow/orange combined image), providing direct chemical proof of TT-dimer formation within the bulk of the condensate. Images are representatives of three different micrographs. The scale bar represents 20 µm for phase-contrast images and 5 µm for confocal images. Source data for panels c and h are provided in the Source data file.

that the mechanical properties of this condensate system are quite robust within the timescale of our study and do not change over tens of minutes.

Following UV treatment, significant changes in the viscoelastic responses were observed (Fig. 3C). In +UV30 samples, G′ and G″ increased substantially compared to control and intersected around 20 Hz, indicating a liquid-to-solid-like crossover. At frequencies below 20 Hz, G′ rose from 5 Pa to ~300 Pa, and G″ from 20 Pa to ~600 Pa. Above 20 Hz, G′ rose more steeply, reaching ~4500 Pa at 100 Hz, while G″ reached ~2600 Pa, reflecting dominant elasticity at high frequencies. In +UV45 samples (Fig. 3D), a similar trend is seen, but the crossover is shifted to higher frequencies ($f > 40$ Hz), where differences between G′ and G″ become nonsignificant, suggesting the

formation of heterogeneous stiff domains. These results support a model in which UV exposure drives condensates from a low-stiffness, liquid-like state to a stiff, solid-like state with slow relaxation at moderate exposure (+UV30), and further UV produces a heterogeneous solid-like system where local stiffening increases elasticity but allows faster relaxation in less-crosslinked regions (+UV45). To better highlight differences between conditions, we also plotted the complex modulus (G*) at 10 Hz (Fig. 3E), revealing a systematic increase in effective viscoelastic stiffness with increasing irradiation duration. Finally, we measured the terminal viscosity of these condensates at $T = 25$ °C by a linear fit of G″ over frequency. Control samples had the lowest viscosity ($6.2 \pm 0.09$ Pa.s), followed by +UV30 ($12.08 \pm 0.11$ Pa.s), while +UV45 condensates showed the highest ($140.77 \pm 11.7$ Pa.s).

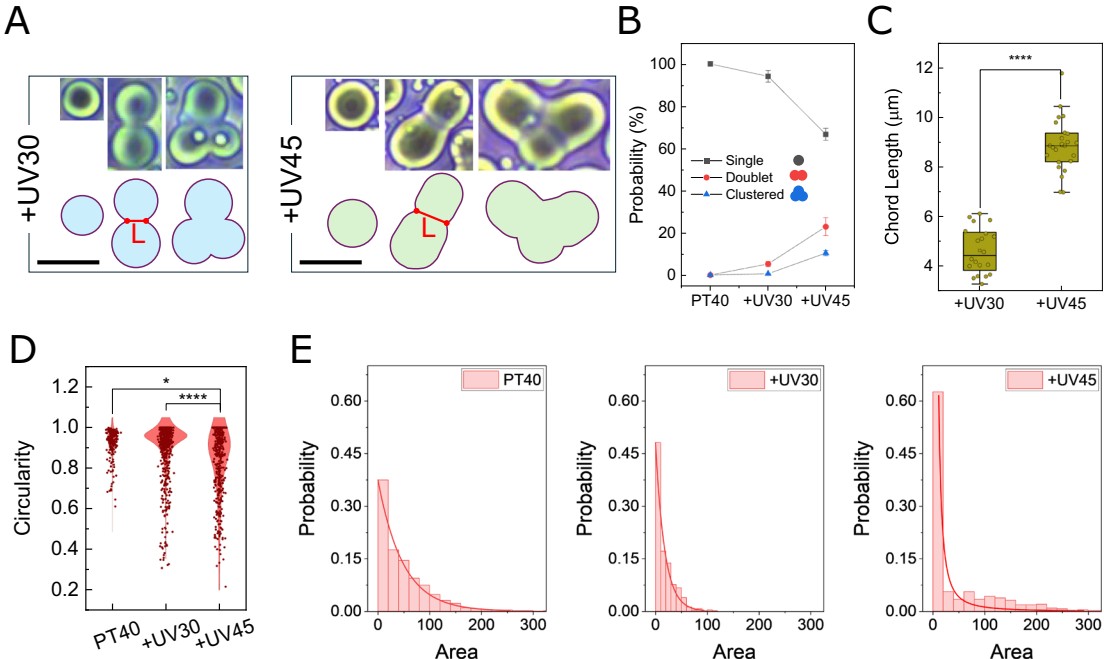

**Fig. 2 | Morphological characterization of PLL:dT40 condensates with and without UV treatment. A** Representative images of single, doublet, and clustered droplets. **B** Comparison of droplet morphologies after UV treatments. The longer UV exposure results in increased probability of doublet and clustered configuration formation. Shape percentages were calculated and averaged across three representative fields of view, with error bars indicating standard deviation. **C** Extended UV treatment increases chord length (L) in doublets. Points represent individual fusion arrested droplets from 3 different micrographs. **D** Morphological analysis of the droplets by calculating the circularity parameter. Values range from 0 (irregular) to 1 (perfectly circular). **E** Size distribution analysis of singlet droplets. Data are fitted with an exponential($a + be^{cx}$) and power-law ($ax^b$) curves. Quantifications were performed on three different micrographs, and error bars represent standard deviation. Box plots show the median (center line), interquartile range (box), and 1.5× IQR whiskers; points represent individual droplets. Statistical significance was assessed using a two-sided Mann–Whitney test; *$p < 0.05$ and ****$p < 0.0001$. The scale bar represents 10 μm. Source data for b–e and exact $p$ values are provided in the Source data file.

## UV irradiation significantly alters nucleation and coalescence processes

UV irradiation induces thymine dimerization in DNA molecules, forming intrachain covalent bonds and, with lower probability, interchain crosslinks[45]. This competition between intra- and inter-chain interactions can significantly influence condensate formation and growth. Conversely, the prevalence of each interaction type may depend on the condensation stage, as condensation itself can modulate photochemical outcomes. To investigate this phenomenon, we conducted UV irradiation for 45 min under two conditions: the "Premix" condition, where UV illumination was applied to dT40 after the addition of buffer and before the addition of PLL, and the "0 h" condition (+UV45_0h), where UV exposure was performed immediately after mixing dT40 and PLL. Interestingly, under the +UV45_0h condition, no condensates were detected by our optical microscopic analysis after 4 h. Instead, a large network of interacting components was formed (Fig. 4A, middle panel). Real-time phase-contrast imaging, however, revealed the transient formation of small condensate-like droplets, which subsequently transitioned into aggregate-like structures during the UV illumination period (Fig. S4A). These structures contained DNA and PLL, as shown by fluorescence analysis (Fig. S4B). Micro-Flow Imaging (MFI) analysis revealed the real-time formation of small particles with a size and circularity index distributions distinct from those of pure condensate systems, suggesting the formation of microaggregates with non-circular geometries (Fig. S4).

Interestingly, in the premix condition, we observed condensate formation (Fig. 4A, left), with smaller sizes (Fig. 4B, C) and circularity values comparable to control droplets (Fig. 4D), indicating no elongation or deformation. Furthermore, premix UV irradiation had minimal impact on condensate mechanics, compared to the control

(Fig. 4E), which may be attributed to the low occurrence of inter-chain interactions. Consistent with this interpretation, molecular exchange analysis showed strongly reduced DNA exchange in +UV45, and efficient exchange in +UV45_Premix and control (PT40, droplets received no UV illumination) droplets. To demonstrate this, we added 1% TYE-665 tagged dT40 to the dilute phase after condensate formation and imaged the droplets after an additional hour of incubation. As shown in Fig. 4F, the +UV45 samples exhibited the lowest fluorescence intensity compared to both the +UV45_Premix and PT40 samples, with control samples displaying the highest intensity (Fig. 4G). Consistent with these observations, FRAP analysis of droplets containing fluorescent components showed rapid DNA fluorescence recovery in both +UV45_Premix and PT40 droplets, but no recovery in +UV45 within the timescale of the experiment (Fig. 4H). Finally, Microscale Thermophoresis (MST) performed on these condensate samples revealed significantly reduced mobility in +UV45, consistent with the formation of inter-chain bonds (Fig. 4I). Together, these findings support a model in which optically regulated interplay between local or long-range intra-chain and inter-chain interactions governs condensate mechanics.

## UV-stabilized droplets exhibit stability and internal compartmentalization

The above observations suggested that UV exposure cross-links DNA molecules in the condensates, thereby altering condensate mechanics and promoting the formation of arrested fusion droplets, which in turn affects chemical compartmentalization. To further assess stability and compartmentalization, we examined the response of +UV45 droplets to extreme changes in the surrounding environment. This was done by replacing the dilute phase with two extreme conditions that, according

to our KCl concentration phase diagram, did not support droplet formation. To do this, following UV irradiation, we removed the original dilute phase and replaced it with pure water (0 M KCl) or an aqueous solution of 1 M KCl. In both cases, the condensates remained stable. In the case of 0 M KCl, phase-contrast microscopy and confocal imaging with FITC-labeled PLL (green) and TYE-665-labeled dT40 (red) revealed the formation of dilute regions within the droplets, indicating an electrostatically driven collapse of the dense network[53] (darker circles inside the droplets in Fig. 5A–D). Washing thus revealed partial compartmentalization induced by UV illumination, which indicates that not all regions were uniformly crosslinked or equally stabilized. Similar dilute compartments have been recently observed in multiple condensate systems upon changes in the ionic strength or temperature[53]. Erkamp et al. showed that this phenomenon occurs when the equilibrium density of the dense phase changes faster than molecular relaxation within the droplet[53]. Specifically, if the droplet contracts faster than the polymer can diffusively redistribute across the droplet, the contraction will open holes of dilute phase within the dense phase. In our system, the removal of salt results in tighter association between dT40 and PLL, favoring contraction of the dense phase network, which occurs much faster than the ~10 min timescale required for molecular redistribution (Fig. 4H).

At 1 M KCl, confocal fluorescence imaging revealed a marked decrease in overall fluorescence intensity compared to droplets treated with pure water (Fig. S5A, B). This reduction indicates a partial release or redistribution of molecular components within the highly screened electrostatic environment. Additionally, unlike the UV-untreated PT40 droplets (Fig. S6), fluorescence intensity line scans across individual droplets revealed an interface with high fluorescence intensity (Fig. S5 C) in both FITC and TYE channels.

## Direct analysis of droplet fusion dynamics using SPM

Our morphological observations indicated that UV-induced thymine dimerization markedly affects droplet fusion dynamics. To directly quantify these effects, we developed an SPM-based assay for condensate droplet fusion and applied it to control (PT40) and +UV45 samples. Figure 6A presents the experimental setup for this measurement. A dual-coated Petri dish with BSA and Pluronic F127 (PF127), as described in detail in the methodology section, was used for making the condensate system. Droplets on the Pluronic-coated surfaces were less sticky and more spherical compared to the BSA-coated side. This approach allowed us to easily capture a droplet, move to the BSA side to approach the target droplet, and perform a droplet-droplet fusion experiment (Fig. 6B).

Our workflow began by selecting a target droplet on the PF127 side and imaging it for diameter determination. The cantilever then approached the droplet under a set force of 0.2 nN (waiting time of 0 seconds), which was sufficient for attachment, thereby capturing the droplet. It was subsequently transferred to the BSA side and positioned adjacent to a selected surface droplet. After imaging, the cantilever's position was adjusted to align the centers of both droplets, and the cantilever was then advanced toward the surface at a constant ("extend") velocity of 1 μm s$^{-1}$ (Fig. 6B), while force–time traces were continuously recorded.

Figure 6C shows a typical force–time measurement recorded during the interaction of two liquid droplets. Initially, the force remained near zero, indicating that the droplets were not yet in contact. As the cantilever moved downward, the force rapidly shifted (starting at $t_1$) to a negative (adhesive) regime when the droplets first came into contact ($t_2$), consistent with the formation of a liquid bridge, although direct imaging of the bridge was not possible with our SPM setup. We refer to the time interval between these two events as the liquid bridge formation time ($\Delta t = t_2 - t_1$). After this initial phase, the cantilever continued to move downward at the same speed as the fusion process proceeded to completion. The (liquid) droplets then merged into a single droplet on the substrate. During the subsequent retract phase, the cantilever moved away from the surface, and the force profile remained in the adhesive regime until the connection between the cantilever and merged droplets ultimately broke. Once detached, the cantilever returned "dry" to its original position, leaving an enlarged droplet on the substrate.

Notably, our data revealed a significant difference in the force-time profiles of UV-treated and untreated samples. In non-crosslinked liquid-like droplets, the force drops instantly and then exhibits a two-stage recovery (Fig. 6D) after reaching the negative minimum: an initial rapid increase due to fast bridge expansion, followed by a slower phase associated with droplet relaxation and substrate wetting. We observed significantly shorter $\Delta t$ (Fig. 6F) and a higher maximum adhesive force (Fig. 6G) in the fusion of control droplets. By contrast, in +UV45 droplets with gel-like behavior, the force decreases gradually, and the distinct two-regime behavior is absent; instead, the force recovery occurs gradually and in a continuous manner (Figs. 6E and S8).

To examine the geometric scaling expected for capillary coalescence, we compared $F_{adhesive, max}$ with the reduced radius $R^* = R_1 R_2/(R_1 + R_2)$ for each droplet pair[54]. Control droplets followed the linear relation $F_{adhesive, max} = 4\pi\gamma R^*$, yielding an apparent interfacial tension (Figs. 6H and S9). UV-treated droplets systematically deviated below this line, which can be attributed to the viscoelastic contributions, not included in the simple capillary model of pure liquids. The observed deviation from the capillary limit can be rationalized by considering that crosslinking introduces an additional mechanical resistance that reduces effective adhesion. To represent these viscoelastic contributions in a compact phenomenological way, we introduced an empirical modulation factor $S(G', G'')$, yielding

$$F_{adhesive, max} \approx \frac{4\pi\gamma R^*}{S(G', G'')}. \quad (1)$$

For un-crosslinked capillary liquids, $S \approx 1$, whereas in viscoelastic droplets $S > 1$ increases with storage and loss moduli. Several functional forms for $S$ reproduce the measured trend, including a rational form

$$S(G', G'') = 1 + a\left(\frac{G^*}{G_0} - 1\right), \quad (2)$$

(with $G^* = \sqrt{G'^2 + G''^2}$) and smooth sigmoidal crossover variants, including a logistic form (Note S2). While not intended as full mechanical models, these phenomenological descriptions capture the continuous transition from a capillary-dominated (surface tension driven) to a viscoelastic-dominated regime in an internally consistent manner. $G_0$ represents the corresponding modulus of the un-crosslinked liquid, providing a dimensionless scaling. The approach is most robust under conditions of slow coalescence, a regime that is characteristic of viscoelastic systems[55]. When applied to other systems, the same framework can be adapted by adjusting parameters to match the characteristic crossover behavior of different condensate chemistries.

We note that the $\gamma$ values calculated from rheology closely agree with those extracted from the capillary model (Fig. 6H) for control droplets; however, the measured values are slightly smaller, which can be attributed to mild facilitation of fusion due to favorable molecular interactions in condensates as compared to pure capillary coalescence (accounted for by $S_0$ in the model, see Note S2). Overall, our observations suggest a model in which the gel-like +UV45 droplets exhibit fusion behavior that is strongly influenced by viscoelasticity (controlling the absolute fusion force), while the heterogeneity in the normalized fusion force reflects contributions from fluctuations in surface tension as well (See Note S3).

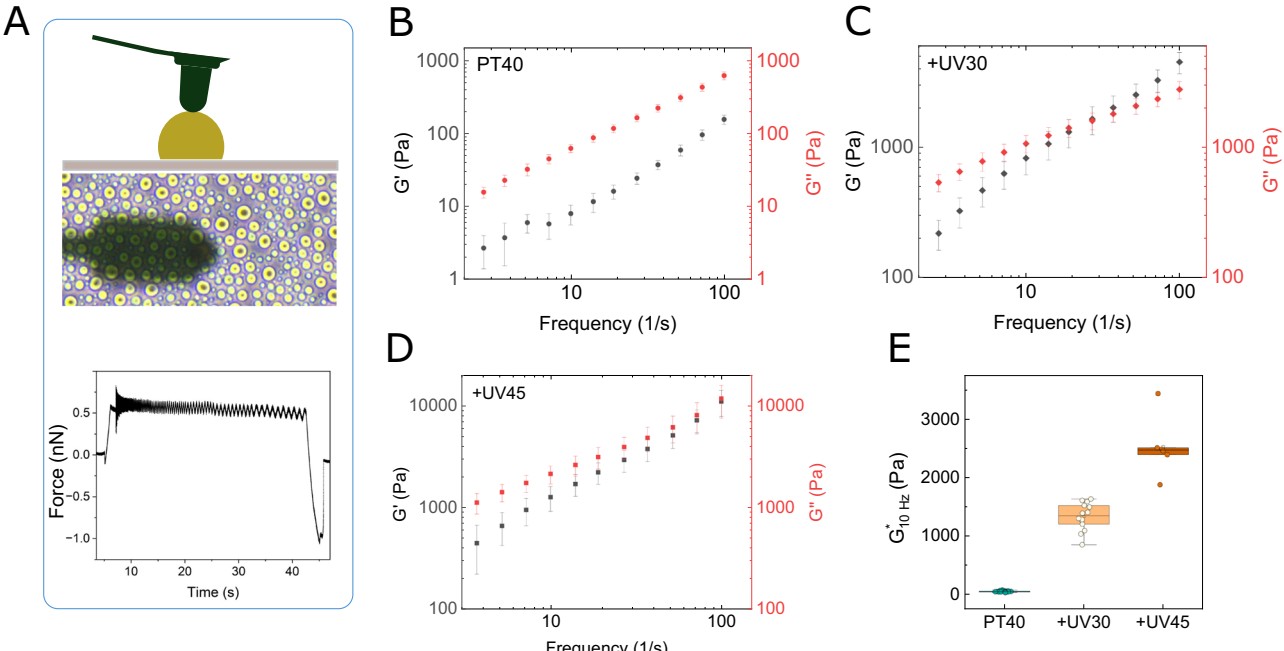

**Fig. 3 | SPM-based mechanical characterization of PLL:dT40 condensates under control and UV-irradiated conditions. A** Schematic of the experimental setup illustrating the scanning probe contacting a droplet on a substrate. **B** Frequency sweep of the storage modulus (G′, black) and loss modulus (G″, red) for the untreated (control) condensate ($n = 49$ different droplets), showing G″ >G′ across the tested frequency range (1 Hz to 100 Hz). **C**, **D** Frequency sweep of G′ (black) and G″ (red) following 30 min (+UV30) and 45 min (+UV45) of UV irradiation, respectively ($n = 14$ and $n = 6$ different droplets respectively). **E** Complex modulus (G*) plotted at 10 Hz for different conditions, demonstrating a progressive increase in viscoelastic response with increasing irradiation time. Points represent individual droplets measured at 10 Hz. Error bars represent standard deviation. Box plots show the median (center line), interquartile range (box), and 1.5× IQR whiskers; points represent individual droplets. Source data for b–e are provided in the Source data file.

Finally, we examined whether the balance between the occurrence of inter- and intra-chain TT dimers affects fusion dynamics. Our rheology analysis showed that in the regime dominated by intra-chain contacts, as seen in +UV45_Premix, both storage and loss moduli are substantially smaller than when inter-chain contacts are frequent, as in the case of +UV45 (Fig. S10). Furthermore, surface tension follows a similar trend, with +UV45 droplets exhibiting higher surface tension. Given that $\phi = \frac{\gamma_{\text{Premix}}}{\gamma_{\text{UV45}}} \times \frac{S(\text{UV45})}{S(\text{Premix})} > 1$, the absolute value of the adhesive force is expected to be higher in +UV45_Premix. Consistent with this, our experimental analysis showed that adhesive force is reduced for +UV45, and the slow structural relaxation of its crosslinked network leads to a marked increase in Δt (Figs. S10 and 7). Moreover, the decrease in Ψ from Premix to +UV45 (observed in both models; See Note S3) is at least 25%, suggesting that accumulation of inter-chain crosslinks suppresses the sensitivity of fusion forces to viscoelastic fluctuations (consistent with the effective saturation of the moduli), thereby shifting the dominant source of variability toward interfacial mechanics.

## Discussion

Mechanics of phase-separated biomolecular condensate droplets can be altered in various ways, including by changing their chemical content, internal structure, or environmental temperature. Here, we showed how light can be leveraged to program the connectivity and internal organization of a nucleic acid-based condensate system, thereby changing condensate mechanics with no alterations to atomic (elemental) composition. UV exposure drives a controlled transition from liquid-like to gel-like droplet states, altering condensate morphology, structural mechanics, and fusion kinetics. This is driven by the formation of intra- and inter-chain bonds that contribute to a marked increase in viscous and elastic moduli, stiffen the droplets towards gel-like/solid-like states, and shift the fusion process from an interfacial tension-driven fast process to a viscoelastic-dominated slow

fusion regime, leading to the observation of arrested fusion droplets. Despite transition towards solid-like states under extreme crosslinking conditions, the system may relax on short time scales, as the organization of the dense domain and surrounding dilute phase enables fast stress relaxation.

Our study presents a concrete example of a photochemical process being modulated by condensation, as the balance between the frequencies of inter- and intra-chain interactions depends on the presence and stages of condensation (Fig. 7). Irradiating the system immediately after mixing the building blocks instead yields a network-like structure[56]. In contrast, applying UV after droplets had formed led to rearrangements primarily within the droplet interior, resulting in condensate mechanical transitions. The premix condition, where the nucleic acid was irradiated prior to adding PLL, produced smaller droplets that retained a largely viscous profile, presumably because intra-chain bond formation dominated in the absence of the polypeptide partner, thus limiting the interaction sites for PLL to form larger complexes and grow. Our observations can be explained by noting that (1) intra-chain bonds bend the DNA, reduce its radius of gyration, and stiffen the polymer, and (2) condensation affects the radius of gyration of polymers as well as the density and inter-chain contact probability[16,57–59], which will in turn regulate the occurrence of short-range and long-range intra-chain and inter-chain thymine dimers due to topological effects and steric constraints.

Our findings provide insights into the role that UV illumination may have played in the origin of life. Early Earth conditions likely exposed primitive protocells to variable doses of solar UV radiation, which could have simultaneously driven molecular crosslinking (through pyrimidine dimerization in early nucleic acids[60,61]) while also posing a threat to nucleic acid stability[62]. Recent models suggest that compartmentalization may have provided a protective niche that shielded nascent genetic material from harmful UV doses[63]. Notably, our findings reveal that UV illumination induces compartmentalization

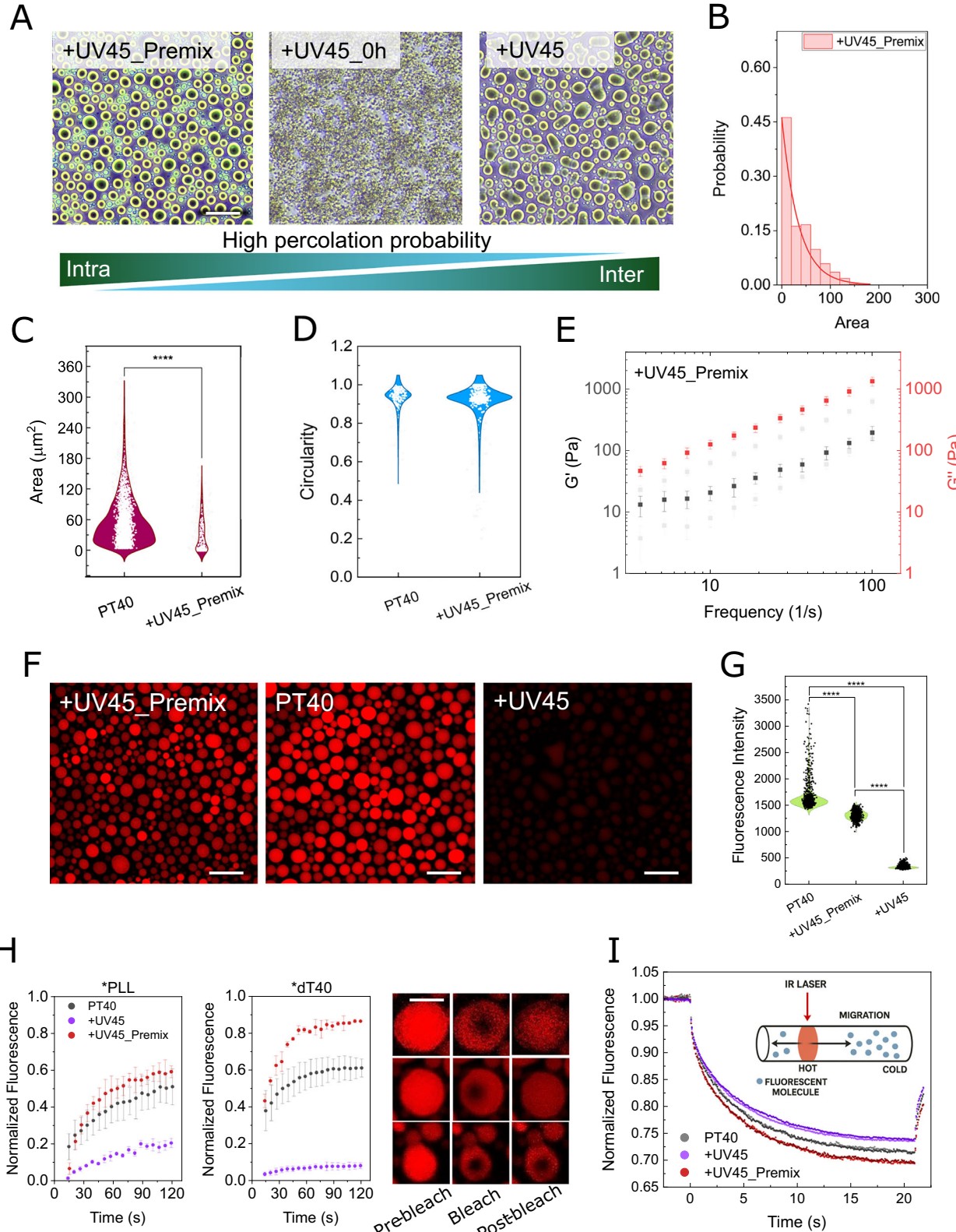

and enhances the stability of the coacervate droplets against drastic changes in environmental chemistry. This mechanism may have played a key role in the resilience and evolution of early biopolymeric systems, balancing the harmful and beneficial effects of radiation on the development of life-like systems.

This study introduces tools and approaches for the synthesis and characterization of materials. The SPM-based assay presented here enables the analysis of condensate droplet fusion dynamics under

physiological and pathological conditions. SPM offers straightforward force calibration, and its open geometry makes it possible to bring separately treated droplets into controlled contact and then characterize the merged object, capabilities that are difficult to achieve with other methods. The proposed viscoelastic modulation model, despite its simplicity, captures the interplay between surface tension and viscoelasticity reasonably well and can be applied to a wide range of crosslinkable condensate systems, although more elaborate

**Fig. 4 | Condensation dramatically impacts the outcome of the UV-induced photochemical reaction in dT40/PLL solution. A** Representative phase contrast images show the results of the three different UV exposure protocols: irradiating the DNA in buffer before adding PLL (+UV45_Premix), applying UV immediately after mixing (+ UV45_0h), or during the final 45 min of a 4 h incubation (+UV45). Control samples with no UV illumination are represented as PT40. Early UV illumination (A, +UV45_0h) yields a percolated network rather than distinct droplets, whereas later irradiation (+UV45) produces elongated but well-defined condensates. **B** Size distribution analysis of droplets in +UV45_premix samples. Data are fitted with an exponential curve. **C, D** Violin plots illustrating the distributions of droplet circularity and size. **E** A frequency sweep of the storage modulus (G′, red) and loss modulus (G″, black)(n = 5 different droplets). This analysis showed no significant differences between control (PT40, light gray) and premix samples.
**F** Confocal fluorescence images from the DNA-exchange assay. After droplet formation, 1% TYE-665–labeled dT40 is added to the dilute phase and imaged after 1 h. +UV45 shows minimal fluorescence, indicating limited DNA exchange into the crosslinked network, whereas +UV45_Premix and PT40 exhibit strong DNA uptake.
**G** Fluorescence intensity analysis of the droplets after addition of the fluorescently tagged dT40 DNA. +UV45 samples showed the lowest intensity, indicating minimal penetration of tagged DNA into the condensate. **H** Dual-channel FRAP analysis of

condensates containing FITC-labeled PLL and TYE-665–labeled dT40 under the three conditions: PT40 (no UV), +UV45_Premix, and +UV45 (n = 3 different droplets). Premix and PT40 droplets exhibited fast fluorescence recovery for both PLL and dT40, whereas +UV45 condensates displayed negligible recovery in either channel over the experimental timescale, indicating severe immobilization of components following UV-induced DNA crosslinking. Representative pre-bleach, bleach, and post-bleach images are represented on the right. **I** Microscale Thermophoresis (MST) characterization of DNA extracted from condensates. Condensates from all conditions were treated with 1 M KCl to dissociate the interaction networks and dissolve the droplets. The released macromolecules from the dilute phase were loaded into MST capillaries, and fluorescence was monitored at the loading location. Samples from +UV45 showed the strongest reduction in thermophoretic mobility, PT40 displayed intermediate behavior, and +UV45_Premix maintained the highest mobility, consistent with the change in the frequencies of inter- (reduced mobility) and intra-chain (enhanced mobility) bond formation. Images are representatives of three different micrographs. Error bars represent standard deviation. Statistical significance was assessed using a two-sided Mann−Whitney test; ****p < 0.0001. The scale bars represent 20 μm for phase-contrast and confocal images and 5 μm for FRAP analysis images. Source data for b−e, g−i and exact p values are provided in the Source data file.

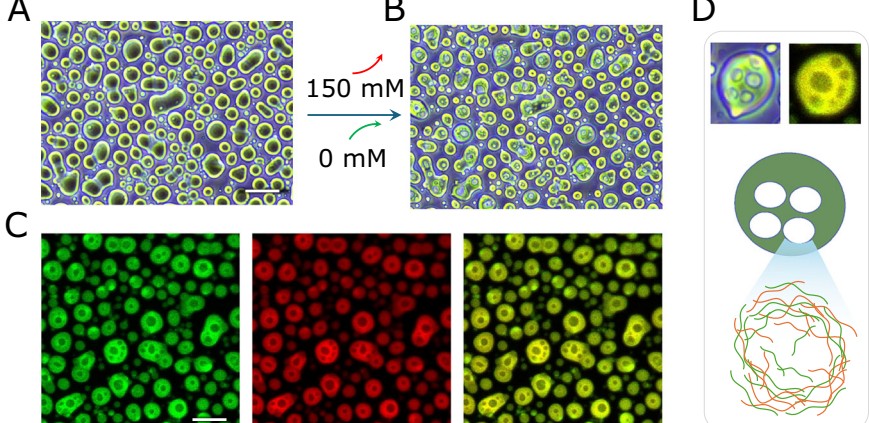

**Fig. 5 | UV-irradiated condensate droplets feature internal compartmentalization and remain stable upon drastic changes to the dilute phase. A** Phase-contrast images showing the morphology of +UV45 droplets before and **B** upon exchanging the dilute phase from 150 mM KCl to 0 M KCl. Droplets remained stable but exhibited internal heterogeneity upon this exchange. **C** Confocal fluorescence

images of +UV45 droplets with labeled FITC-PLL (green) and TYE665-DNA (red) in 0 M KCl. **D** Magnified view and schematic illustration of a representative droplet, highlighting the observed internal compartmentalization. Images are representatives of three different micrographs. The scale bars represent 20 μm for phase-contrast and confocal images.

phenomenological forms (e.g., multi-parameter crossover) and models with explicit microscopic grounding could be considered in more complex scenarios. Furthermore, the synthesis of stabilized condensate droplets with tunable mechanical properties may open new avenues in materials and particle synthesis for biomedical and engineering applications. Such tunability offers opportunities for the design of programmable biomolecular materials, especially given the contact-free nature and experimental convenience of UV-induced crosslinking[64]. The approach presented in this study is versatile and can be readily applied to other polyelectrolyte−nucleic acid systems or under different environmental conditions. It thus opens avenues for engineering synthetic organelles and soft materials that dynamically respond to changes in their environment.

## Methods
### Preparation of ssDNA and Poly-L-Lysine Stock Solutions
ssDNA oligo of dT40 was purchased from Integrated DNA Technologies (NJ, USA). Dry stocks were reconstituted in RNase-free water, without added salt, and DNA concentration was measured subsequently using a Qubit spectrophotometer. The DNA stocks were then aliquoted and stored at −20 °C until use. Poly-L-lysine (PLL) hydrochloride with an average molecular weight of 15,000 to

30,000 Da was used in all experiments. The PLL was dissolved in water to prepare stock solutions at 22.4 mg/ml. Based on the average molecular weight, PLL is ~150 residues long, representing a simple model for the diversity of positively charged polypeptides in cells or primitive life forms.

### Surface passivation for rheological and droplet-droplet interaction analysis
Condensate samples for phase-contrast imaging, mechanical measurements, and droplet-droplet fusion assays were prepared on TPP Petri dishes. For rheology, dishes were passivated with 1% w/v BSA (A2153) for 30 min and washed five to six times with RNase-free water.

For droplet-droplet fusion, droplets needed to be detached from the substrate with the cantilever and positioned onto target droplets. To enable this, the dish was half-coated with Pluronic and half with BSA: Pluronic provided weak adhesion for easy droplet detachment, whereas BSA promoted stronger adhesion, ensuring stable coalescence during fusion. A guideline was drawn on the underside of the dish to ensure consistent passivation. The dish was fixed in position and tilted ~20°. For each passivation, the bottom half of the dish was filled to the guideline, incubated for 30 min, and washed multiple times with RNase-free water.

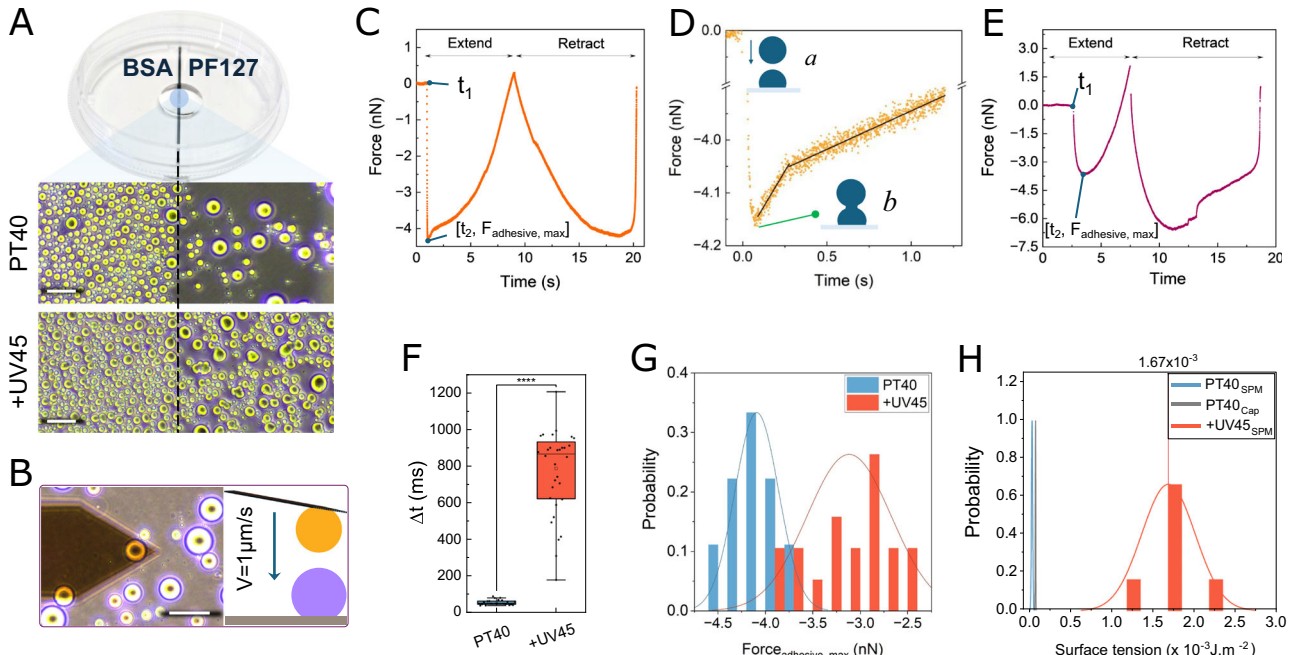

**Fig. 6 | SPM-based droplet–droplet interaction analysis reveals UV-induced changes in fusion dynamics. A** Dual-area coated dish for condensate preparation. Comparison of droplet behavior on a dual-coated Petri dish with BSA (left) and Pluronic PF127 (PF127; right) surfaces. The black line divides the two coated regions. Droplets on the PF127 side are more rounded and exhibit reduced stickiness compared to the BSA side, where droplets adhere more strongly to the surface (shown in the lower panels). **B** Experimental setup for droplet-droplet interaction analysis using SPM. **C** Representative example of the recorded Force-time curve for liquid-like PT40 droplets. A cycle includes an extend phase and a retraction phase. During the extend, the piezoelectric actuator is extended, moving the cantilever tip toward the sample surface. The cantilever, with an attached droplet, was moved downward (Extend phase) at $1\,\mu m\,s^{-1}$ until it contacted the droplet on the substrate, consistent with a liquid bridge formation. Following initial contact, the cantilever continued to move downward as droplet fusion progressed to completion, after which it retracted to its original position. The negative portion of the force curve indicates capillary attraction (adhesion) and shows dependency on approach velocity (Fig. S7). **D** Zoomed in force–time trace showing the interactions between two liquid-like PT40 droplets. The force

remains near zero before contact (a), rapidly drops to a negative minimum, suggesting bridge formation (b), and then transitions through two distinct regimes of coalescence and droplet reshaping. **E** Representative force-time trace for +UV45 sample. **F** Box plot comparing the bridge formation times ($\Delta t = t_2 - t_1$) for PT40 ($n = 19$ different pairs of droplets) and +UV45 samples ($n = 16$ different pairs of droplets). The +UV45 sample shows a significantly larger $\Delta t$. **G** Maximum adhesive force ($F_{adhesive,\ max}$) during droplet interaction for PT40 and +UV45. The +UV45 sample shows a shift toward lower adhesive forces, suggesting that photo-crosslinking alters the capillary interactions between droplets by increasing structural resistance to deformation. **H** Surface tension ($\gamma$) extracted from force–time traces for PT40 based on the capillary model ($n = 17$ different pairs of droplets), and from SPM analysis for PT40 ($n = 15$ different droplets) and +UV45 samples ($n = 6$ different droplets). Box plots show the median (center line), interquartile range (box), and 1.5× IQR whiskers; points represent individual droplets. Statistical significance was assessed using a two-sided Mann–Whitney $U$ test; ****$p < 0.0001$. The scale bars represent 20 μm. Source data for c–h and exact $p$ values are provided in the Source data file.

## Sample preparation

For each experiment, 60 μL of the working buffer (150 mM KCl (p9333), 10 mM imidazole (56750), and 0.01% w/v NaN$_3$ (8.22335))[65] was placed as a round droplet at the center of the dish. For droplet-droplet analysis, this step was performed carefully to align the droplet's middle axis with the drawn guideline. To mimic a crowded biological environment, the crowding agent Ficoll® PM 70 (F2878) was added to the buffer at a final concentration of 50 g/L. Subsequently, ssDNA and poly-L-lysine (PLL, P2658) were added sequentially to reach a final concentration of 30 μM for each. After adding each component, the solution was gently mixed by pipetting 10 times.

## SPM measurements

A JPK CellHesion 200 (Bruker, Germany) equipped with a phase contrast microscope delivering real-time in-situ images of measurements utilizing a 20X/0.4 objective was employed in this study. The system spans the piconewton to micronewton regime, depending on cantilever stiffness, and provides an extended vertical range of motion, enabling mechanical interrogation of biomolecular condensates across molecular to mesoscale forces. Here, mechanical analysis was performed as explained by Naghilou et al.[22]. Briefly, a 60 μl bulk drop

containing the condensate components was prepared, and the condensate droplets were allowed to settle on the dish. All measurements were performed within 4 h of droplet formation, unless stated otherwise. Before each mechanical measurement, a 5 μl drop was carefully taken from the top of the bulk droplet and placed on the cantilever to avoid the generation of air bubbles when the head is placed. Micrometer-scale indenters were preferred over nano-scale counterparts for improved signal-to-noise ratio[66]. For all experiments, SAA-SPH-5UM cantilevers (Bruker, Germany) made of Si$_3$N$_4$ with a hemispherical tip (23 μm height, 5.13 μm radius) were used. The calibration of the amplitude and the exact cantilever spring constant was determined with the thermal noise method[67]. The cantilever was passivated with 1% Pluronic F127 (P2443) for 30 min to prevent the adhesion of droplets on the tip after measurements[68]. Condensates (preferably of 8-10 μm in diameter) were indented with a 0.3 nN set force and a cantilever approaching with a constant velocity of 1 μm/s. Acquired data was post-processed by Bruker data analysis software and prepared for calculation of rheological parameters using custom Mathematica code (Mathematica 14.0, Wolfram). Size distribution analysis and calculation of the radius of condensates were done using ImageJ (https://imagej.nih.gov/) from the phase contrast images.

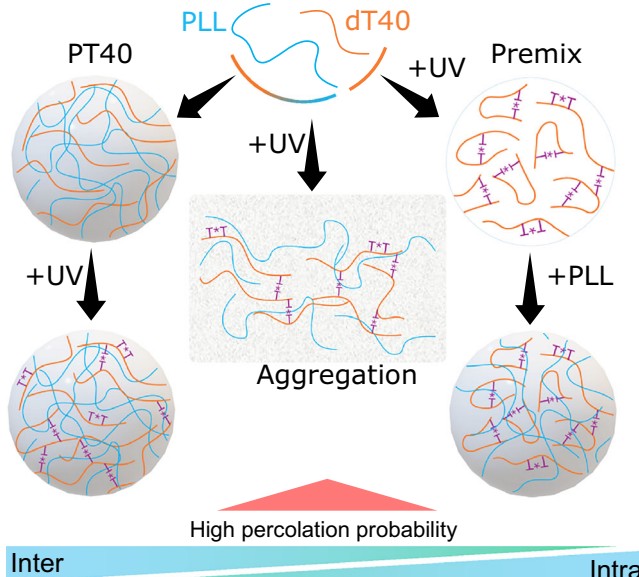

**Fig. 7 | Reciprocal interplay between photochemical reactions and condensation, governed by the balance between inter- and intrachain interactions.** Our data indicate the dominant role of the intra-molecular interactions in premix conditions, which decreases in samples exposed to UV after condensates formed, mainly due to the presence of interacting partners in the environment and increases in the dT40 gyration radius, which favor inter-molecular interactions. Between the two extremes, UV irradiation leads to the formation of a percolated network of interacting polymers in +UV45_0h condition, which results in the formation of aggregates with globally connected chains. The red triangle denotes an increased probability of network percolation in +UV45_0h UV-illuminated samples.

## Confocal fluorescence imaging

Confocal fluorescence imaging was performed using a Nikon Ti2 inverted microscope with a Nikon C2plus confocal system and a Plan Apo VC 20x DIC N2 objective (NA = 0.75, WD = 1000 μm). Imaging was conducted with a Galvano scanner in one-way scan mode at 6.897x zoom. Excitation was achieved with 489 nm and 562 nm lasers for GFP and TYE-665, respectively, with emissions detected at 540 nm and 665 nm. The detector gain was set to 100 (GFP) and 70 (TYE-665), and the pinhole size was 40 μm. A brightfield transmission detection (TD) channel was also acquired. Images (512 × 512 pixels) were acquired in three planes: GFP, TYE-665, and TD, with an exposure time of 1095.456 ms at a voxel size of $1 \times 1 \times 1$ pixel$^3$. Image processing and analysis were performed using Fiji (ImageJ) version 2.14.0/1. TIFF-formatted images were used for quantitative and structural analysis without additional thresholding or scaling.

## UV−visible spectroscopy

UV−visible spectroscopy was performed to evaluate photochemical modifications to the dT40 oligonucleotide following UV irradiation. A final concentration of 30 μM dT40 was used for all experiments. For the +UV45_Premix samples, the DNA was mixed into 60 μl of working buffer and irradiated for 45 min, after which a sample was taken for spectroscopic analysis. For the +UV45 samples, 45 min of irradiation were applied after 3 h and 15 min of the condensate formation period. Subsequently, high salt concentration was added to the droplets to release the dT40 into the dilute phase, from which a sample was collected. Non-irradiated control samples were prepared by mixing dT40 in the working buffer and analysed immediately. Absorption spectra were recorded using an Agilent 8453 UV−Vis spectrophotometer (Agilent Technologies, Santa Clara, CA, USA) in a low-volume quartz cuvette across the standard nucleic-acid absorption range. The resulting spectral changes were used to identify UV-induced TT dimerization in dT40.

## Microfluidic imaging (MFI)

Particle imaging was performed using a ProteinSimple MFI system (ProteinSimple, San Jose, CA, USA) operated in basic mode using default acquisition parameters with a 200 μl/min flow rate. Particle identification and quantification, including particle counts, equivalent circular diameter (ECD), and circularity, were carried out with the instrument's analysis software (MFI View System Software, ProteinSimple, San Jose, CA, USA). Processed datasets were exported and plotted using OriginLab 2025 (OriginLab Corporation, Northampton, MA, USA). The approach was applied to +UV45_0h samples, right after UV illumination, and the size and morphology of the floating objects were quantified.

## Fluorescence recovery after photobleaching (FRAP)

FRAP experiments were performed to quantify the relative recovery dynamics of PLL and DNA within individual condensate droplets. To enable simultaneous measurement of both components, 5% FITC-labeled PLL (P3543; Sigma-Aldrich, St. Louis, MO, USA) was mixed with its unlabeled counterpart, and 5% TYE-665–labeled dT40 was incorporated into the DNA component. Sample preparation followed the procedures described above.

All measurements were carried out on a Nikon Eclipse Ti2 microscope (Nikon Instruments, Tokyo, Japan) equipped with a 20×/0.75 NA objective. A circular region of interest (radius 1.25 μm) within each droplet was bleached for 1 s using a 488 nm laser at 250 μW, ensuring that droplet diameters were at least two to three times larger than the bleach radius. Fluorescence recovery was recorded for 3 min at low excitation power to minimize additional photobleaching. Both fluorescence channels (FITC for PLL and TYE-665 for DNA) were acquired sequentially for each droplet. Fluorescence intensities were extracted from the bleached region ($F_{ROI}$), an unbleached reference droplet ($F_{ref}$), and a background region ($F_{bkgd}$). To correct for photodecay, intensity traces were normalized to the reference droplet according to:

$$F_{norm}(t) = \frac{F_{ROI}(t) - F_{bkgd}(t)}{F_{ref}(t) - F_{bkgd}(t)} \cdot \frac{F_{ref, i} - F_{bkgd, i}}{F_{ROI, i} - F_{bkgd, i}} \quad (3)$$

where $F_i$ denotes the pre-bleach average intensity. Normalized recovery curves were used to compare relative recovery times of PLL and DNA within the same droplets.

## Microscale thermophoresis

Microscale thermophoresis (MST) was used to assess the mobility of dT40 molecules in PT40, +UV45_Premix, and +UV45 samples. All samples were prepared at a final dT40 concentration of 30 μM. For the +UV45_Premix condition, dT40 was added to 60 μL of the bulk solution, irradiated for 45 min, and samples were collected immediately after irradiation. For the +UV45 condition, droplets were allowed to form for 3 h 15 min before 45 min of UV exposure. High-salt buffer was then added to dissolve the condensates and release dT40 into the dilute phase, from which material was collected. Non-irradiated controls were prepared by mixing dT40 directly into the working buffer and measured immediately.

MST measurements were performed on a Monolith NT.115 instrument (NanoTemper Technologies, Munich, Germany) using standard capillaries. Samples were equilibrated for 10 min at room temperature before loading. Fluorescently labeled DNA (5% TYE-665 dT40) was mixed with unlabeled samples. Thermophoretic traces were recorded at 25 °C using medium IR-laser power. Normalized fluorescence ($F_{norm}$) was extracted using the manufacturer's software, and changes in $F_{norm}$ were used to quantify differences in ssDNA mobility associated with UV-induced TT dimer formation.

## Anti-CPD immunofluorescence staining assay

Cyclobutane pyrimidine dimer (CPD) formation within condensates was assessed by immunofluorescence staining using a FITC-conjugated anti-CPD antibody from the OxiSelect™ Cellular UV-Induced DNA Damage Staining Kit (CPD, Cell Biolabs). Condensate samples contained 5% TYE-665–labeled DNA (of the total DNA) and were incubated with the anti-CPD primary antibody diluted 1:100 in assay diluent for 1 h at room temperature with gentle agitation. After three washes with 1× wash buffer, droplets were incubated with a FITC-conjugated secondary antibody (1:100 in assay diluent) for 1 h under identical conditions. Excess antibody was removed by four additional washes with 1× wash buffer. Fluorescence microscopy was performed using channels appropriate for both TYE-665 and FITC fluorescence detection.

## Reporting summary

Further information on research design is available in the Nature Portfolio Reporting Summary linked to this article.

## Data availability

All the data supporting the findings of this study are available within the paper, the supplementary information files and the Source Data files (SourceData and SourceData_SI). The unprocessed data for the work is also available from the authors upon request. Source data are provided with this paper.

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

## Acknowledgements

This project was partly supported by the Dutch Research Council (NWO), Open Competition grant OCENW.XS23.3.105 (A.M.). The authors thank Sander Woutersen (University of Amsterdam) for valuable discussions. The authors also gratefully acknowledge the support of the Leiden Cell Observatory (LCO). In particular, we thank Kostas Tassis of the LCO for his assistance with the early phases of confocal microscopy.

## Author contributions

Conceptualization, A.M. and V.S.; methodology, A.M. and V.S.; investigation, V.S., A.M., D.B., J.D.S.; visualization, V.S.; formal analysis, V.S. and F.H.K.W.; writing original draft, V.S. and A.M.; writing review & editing, V.S., A.M., J.D.S., D.B., F.H.K.W.; project administration, funding acquisition, and supervision, A.M.

## Competing interests

The authors declare no competing interests.
