## [Transparent Peer Review file · Nature Communications]

Optically driven control of mechanochemistry and fusion dynamics of biomolecular condensates via thymine dimerization

Corresponding Author: Professor Alireza Mashaghi

Version 0:

Reviewer comments:

Reviewer #1

(Remarks to the Author)

In this manuscript, Sheikhhassani et al. studies how UV-induced photochemical reactions alter the mechanics of ssDNA containing (PLL/dT40) condensates. The authors found that UV illumination leads to solidification of condensates, likely through thymine dimerization. They also observed the formation of internal compartments and arrested fusion of condensates.

Overall, the paper is well organized and easy to read. However, while several observations are interesting, most data are qualitative (with the exception of Figure 2) and lack molecular insights. Additionally, the biological relevance of the current paper appears limited. Therefore, we believe a properly revised version of the manuscript is better suited for a more specialized journal.

Our comments are listed below:

1. The authors used the scanning probe microscopy (SPM) method developed in a previous study. However, comparison between SPM and existing tools (OT, micropipette aspiration) should be discussed more objectively. Importantly, it's unclear why the authors claim SPM to have 'minimal perturbation' and 'more straightforward to integrate with UV illumination' compared to other assays.
2. In SPM measurements, it's unclear if passivation is necessary and if it will affect the viscoelasticity quantifications.
3. It's important to provide the power density of the UV illumination.
4. The extensive use of condensate morphology (eg, circularity, semi-fusion, elongated shape) is concerning. This is because A) the morphology of condensates are strongly affected by the adhesion between condensates and the substrate (this adhesion is clearly not removed for most experiments). For example, see (<https://www.freeimages.com/premium/water-droplets-on-a-metal-surface-1592483>). B) condensates with similar circularity can have very different mechanical properties, as illustrated in several recent studies (for example, <https://pubmed.ncbi.nlm.nih.gov/39211102/>).
5. The authors should better explain the size distribution of condensates and their corresponding formation mechanisms. These are unlikely common knowledge for the readers. Along this line, how does condensate surface tension affect its size distribution?
6. In figure 2, the observation that the crossing point between G' and G'' increase between 30 and 45 min UV is surprising. How does this reconcile with the solidification of the condensate and the observed change in terminal viscosity (which itself needs to be better defined)?
7. Does aging play a role in the liquid to solid transition of the PT40 condensates? For example, it would be better to include a set of data a separate plot to show the material properties of PT40 is stable for at least 45 min.
8. A major point of the paper is thymine dimerization induced by UV. While plausible, we didn't find any direct evidence in the paper to support this claim. Could the authors check with electrophoresis?
9. The observation of compartments within condensates is interesting. But this result is very descriptive and lack of mechanistic insights.
10. The fusion between two condensates is affected by condensate viscosity. So, it's unclear if data in figure 5 can be used to draw direct conclusion on condensate surface tension and if it is affected by the speed of the measurement. Plus, the

claim of 'bridge formation' does not appear to be directly supported by imaging evidence.

Minor problems:

1. Line 82 'preserving chemical composition' is inaccurate since dimerization is a chemical change.
2. In line 167, the author mentioned "this condition produced droplets of convenient size for mechanical analysis" but did not state what is the actual size in the main text or main figures.
3. Figure 1A and many other images are missing scale bar.
4. Figure 1F. I would recommend defining what is single/paired/cluster with images.
5. Line 232, unclear if the values were quoted for a specific frequency.
6. Line 246, reference missing.
7. The description and alignment of Figure 3A and 3B are overall confusing. Since A and B are two different experiments. It is better to make separate text descriptions for them under figure 3.
8. Although "applying UV immediately after mixing (+UV45_0h)" was observed with no condensate formation in 3A, it still recommended to include a confocal image of +UV45_0h with 1% TYE-665 tagged dT40 in Figure 3B.
9. Figure 3F meant to show no significant differences between control and premix samples but the control is not plotted.
10. Line 394, the use of reduced radius is unclear.

Reviewer #2

(Remarks to the Author)

Reviewer #3

(Remarks to the Author)

In the submitted manuscript, scanning probe microscopy is combined with optical microscopy to study the mechanical properties and fusion dynamics of condensate droplets. The droplets are formed in solutions of two polyelectrolytes, single-stranded DNA consisting of 40 thymine nucleotides (oligo-ssDNA dT40), which carries 40 negative charges, and the protein PLL with an average length of 150 residues, carrying 150 positive charges. The solution undergoes phase separation within a certain concentration regime as characterized in Figure S1.

When the polyelectrolyte solution is illuminated with UV light, the two polymers form crosslinks which can drive the droplets from a liquid to a gel-like or solid-like state, thereby changing the mechanical properties of the droplets and their morphology. The transition from the liquid to a gel- or solid-like state can be used, for instance, to arrest the fusion of the condensate droplets in some intermediate droplet state with nonspherical morphology as shown in Figure 1G.

My main concern is that the results reported in this manuscript are quite reasonable from an intuitive and qualitative point of view whereas the analysis of these results is rather fragmented, often confusing, and difficult to follow.

I recommend major revision of the submitted manuscript. This revision should, in particular, address the following points, ordered in the same way as the corresponding manuscript pieces in the main text and in the supplementary information:

1) The title of the manuscript emphasizes thymine dimerization and the abstract claims on page 1, line 27 and 28, that "UV-induced thymine dimerization alters condensate nucleation and coalescence". However, this claim remains obscure until Figure 6 at the end of the manuscript. Therefore, I urge the authors to mention the competition between covalent intra- and interchain contacts, which is currently explained in Figure 6, already in the abstract and to further clarify the text pieces before and after Figure 6.

2) At the end of the abstract, the authors claim that their findings have "implications for genome biology, the emergence of life on Earth, and engineering applications."
I can see possible engineering applications, related, for instance, to synthetic biology but I have severe problems to imagine

implications for genome biology and for the emergence of life on Earth. In fact, "genome biology" is not mentioned anywhere else in the manuscript and the "origin of life" is only briefly mentioned in line 446 on page 17 at the very end of the manuscript.

If the authors want to maintain these claims, they should tell the interested reader in much more detail what they have in mind.

3) Related to point 2): Condensate droplets in living cells have been discovered in 2009 and have been intensely studied during the last two decades by a variety of experimental and theoretical methods. It is now understood that intracellular condensate droplets are usually in a liquid-like rather than in a gel-like or solid-like state. Therefore, do the authors really believe that gel-like or solid-like condensate droplets play some important role in living cells? If yes, they should explicitly describe their line of reasoning in the manuscript.

4) The first paragraph of "Materials and Methods" provides some information about the preparation of the stock solutions. This information is, however, not sufficient if someone wants to repeat this preparation.

As far as dT40 is concerned, one would like to know about the buffer used to rehydrate the dry dT40 obtained from the producer. In particular, which salt was added at what concentration? And was the pH controlled during this preparation?

As far as the PLL protein is concerned, I would have expected that this protein was produced by some recombinant method, which would lead to a well-defined length of the protein.

However, in line 101 and 102 on page 4, the authors mention the "average" molecular weight and the "average" length of the PLL protein. Where does the polydispersity of PLL come from and how large is it?

5) After cross-linking the two polymers by UV illumination, the droplets exhibit an increased viscosity as reported in Figure 2E. This increase is dramatically different depending on the protocol used for the illumination. The lettering at the x-axis of Figure 2E distinguishes three cases: "control", "30minUV:4hrs", and "45minUV:4hrs".

One might think that "xyminUV:4hs" stands for "applying UV for xy min after 4 hours", but these abbreviations are not explained in the text. Instead, in line 241 on page 9, the same protocols are called "+UV30" and "+UV45". Furthermore, in line 249 on page 9, the authors say that the UV irradiation was applied "for 45 minutes" without mentioning the alternative exposure time of 30 minutes. In addition, the authors now refer to the "Premix" condition and to the "0 hours" condition.

Therefore, the description of the different protocols as used in the submitted manuscript is very confusing and must be improved. The same comment applies to Figure 3.

6) Related to point 5): One general advantage of light-induced changes of molecules and chemical reactions is that these changes can be rather fast. Indeed, the publication by Johnson and Wiest, JPC B (2007), Ref 11 of the manuscript, emphasizes that "thymine dimerization is an extraordinarily rapid reaction." In contrast, the time scales discussed in the submitted manuscript for the UV illumination protocols are very long, corresponding to 30 or 45 minutes.

Thus, the authors should explain why the UV-induced dimerization studied here is so much slower.

7) The results for the apparent interfacial tension are displayed in Figure 5H in the form of a histogram. This histogram exhibits at least two, maybe even three peaks. The authors should explain whether or not they can interpret this histogram in terms of the underlying molecular processes.

Does this histogram reflect, for instance, the competition between

covalent intra- and interchain contacts of the polymer chains?

8) Two typos on page 15: in line 399, Figure 5H should be replaced by Figure 5G, and in line 402, Figure 5I should be replaced by Figure 5H.

9) Figure S1 is quite useful in order to understand the formation of the biomolecular condensates studied here, Therefore, the authors might want to move Figure S1 into the main text.

Reviewer #4

(Remarks to the Author)

In a nutshell, the paper reports that UV illumination can alter the fusion times of condensate droplets. I haven't studied the literature exhaustively, but this does seem to me like a new and interesting discovery. I have comments:

- I am not an expert in this specific area, but the authors state with great confidence that the observed changes in condensate mechanics are due to thymine dimerization. I do not see direct evidence for this claim; at present it seems more like a plausible hypothesis rather than a demonstrated molecular mechanism.
- In Fig. 1G and H the authors analyze "neck length." While this is a reasonable empirical observation, is there a quantitative framework that could connect this parameter to rheological or other mechanical quantities?
- I find the introduction poorly written and the results written with a somewhat overstated style. For example:
 - o Micropipette aspiration does not, in general, require the use of tracer particles.
 - o To my knowledge, thymine dimerization involves the formation of covalent bonds, which necessarily alters the chemical composition. Therefore, the sentence claiming that chemical composition remains unchanged should be revised.
 - o The text contains multiple "first-of-its-kind" statements that are unnecessary.
 - o Prior analogous work should be acknowledged, for instance the use of AFM to measure the material properties of objects as large as condensates (<https://www.nature.com/articles/nphys4104>).
 - o The paper is about material properties and their relation to molecular conformations, but does not cite the significant very recent progress in this area from the Banerjee, Mittal, and Schuler groups.
- Line 332 seems to be referring to Figure 5B. Please check.
- Finally, I find it puzzling that no condensates are reported in the "0h" condition. Could it be that phase separation does occur, but the droplets are arrested at sizes below the diffraction limit of the imaging system?

Version 1:

Reviewer comments:

Reviewer #1

(Remarks to the Author)

The authors have largely addressed our concerns in the revised manuscript. We only have the following suggestions:

1. Figure 1 in the response letter show a ~5 fold difference between two coating conditions. The data do not exclude the possibility the coating can have a significant effect the measured viscoelasticity of condensates. We think the authors should be more cautious about this point and include this result as a supporting information in the manuscript.
2. We were unable to locate the power density values of the UV illumination.
3. The authors claim that SPM has higher range of forces compare to OT and micropipette aspiration without reporting the force range of SPM. But this is not supported by any direct evidence.
4. The number of samples and independent measurements are missing in several figures, including figure 2 and s2.

Reviewer #2

(Remarks to the Author)

Reviewer #3

(Remarks to the Author)

The authors have responded to all concerns raised in my previous report and made appropriate changes in the revised manuscript. The manuscript is now suitable for publication in Nature Communications.

Reviewer #4

(Remarks to the Author)

Good for publication.

REVIEWER COMMENTS

We are grateful to the referees for their critical feedback and suggestions, which helped us improve the manuscript. Below, we provide point-by-point responses to their comments and describe the revisions made.

Reviewer #1 (Remarks to the Author):

In this manuscript, Sheikhhassani et al. studies how UV-induced photochemical reactions alter the mechanics of ssDNA containing (PLL/dT40) condensates. The authors found that UV illumination leads to solidification of condensates, likely through thymine dimerization. They also observed the formation of internal compartments and arrested fusion of condensates.

Overall, the paper is well organized and easy to read. However, while several observations are interesting, most data are qualitative (with the exception of Figure 2) and lack molecular insights. Additionally, the biological relevance of the current paper appears limited. Therefore, we believe a properly revised version of the manuscript is better suited for a more specialized journal.

Response:

Thanks for your positive remarks on the structure and readability of the paper and the constructive comments. As described below, we made extensive revisions to the paper and added substantial new data to better highlight the findings and impact of our work on the field of biomolecular condensates. Importantly, we have addressed the reviewer's concern regarding the qualitative nature of the data and underlying molecular mechanisms, which we elaborate on in the remainder of this rebuttal. The work reveals how the balance of inter- versus intrachain contacts governs condensate mechanics and fusion, providing key insights into biomolecular phase behavior.

Our work presents unprecedented insights into the control and engineering of condensate microdroplets and introduces powerful tools and model systems for investigating condensate mechanochemistry. These technological innovations enabled us to make the following key discoveries:

- 1) Chemistry affects condensation:** Our study shows that the simplest chemical reaction involving a photochemical-induced formation of a covalent bond affects condensation without changing the chemical content of the system.
- 2) Condensation affects chemistry:** We show that the outcome of a photochemical reaction is tightly dependent on the presence or absence of condensation.

Moreover, the demonstrated ability to control condensate phase and mechanochemistry, as well as to synthesize compartmentalized condensate particles, provides new bio-inspired engineering approaches relevant to chemical engineering and materials science.

Therefore, we believe that our study offers fundamental insights into the relationship between condensation and chemical reactivity, while also introducing unique, versatile tools for the precise engineering of condensate properties. This work establishes biomolecular condensates as a chemically programmable system, linking their mechanochemistry to reaction control and material design, with broad implications for chemistry beyond biology. While we have revised the manuscript to avoid overstating immediate biological relevance, we emphasize that the technological contributions are expected to have a significant impact on both chemical and biological research. Given its broad implications, we are of the opinion that our work will be of great interest to the readers of Nature Communications.

Our comments are listed below:

1. The authors used the scanning probe microscopy (SPM) method developed in a previous study. However, comparison between SPM and existing tools (OT, micropipette aspiration) should be discussed more objectively. Importantly, it's unclear why the authors claim SPM to have 'minimal perturbation' and 'more straightforward to integrate with UV illumination' compared to other assays.

Response:

Thanks for this comment. In the original version, our description of SPM included statements such as "minimal perturbation" and "more straightforward to integrate with UV illumination," which were unnecessary and could be interpreted as overclaims. We have removed these phrases and rewritten the section to avoid implying advantages that are not essential to our argument. SPM does provide a convenient way of performing these experiments due to the easy integration of a UV treatment module. While such integration is theoretically possible with custom-built optical tweezers, current commercial systems do not offer a practical way to achieve this.

We have fully rewritten the corresponding introductory paragraph and added references to our recent studies, published while this manuscript was under review, where we established the use of SPM for condensate rheology analysis. In these papers, comparisons with other techniques are discussed objectively, including the relevant force regimes and whether tracer particles or exogenous probes are required.

Finally, we emphasize in the revision that the use of SPM for direct droplet–droplet fusion measurements is new to this work. In contrast to optical tweezers, where force calibration is challenging in condensate fusion studies, SPM allows straightforward calibration. Moreover, the open geometry and ability to mechanically manipulate droplets make it feasible to bring separately treated droplets into controlled contact and subsequently characterize the merged object capabilities that are difficult to implement in closed OT chambers. Specific droplets can then be subjected to downstream analysis using analytical methods.

2. In SPM measurements, it's unclear if passivation is necessary and if it will affect the viscoelasticity quantifications.

Response:

Thanks for this comment.

Passivation is indeed necessary for SPM analysis of these condensates to ensure the formation of round droplets. Making condensates on uncoated SPM dishes typically leads to the spreading of the condensates due to wetting. The reviewer asks a valid question on whether passivation has any impact on viscoelasticity quantifications. While passivation is commonly used in biomolecular condensate research, we agree that this aspect merits careful investigation. To address this, we performed new sets of experiments in which two very distinct passivation protocols were used, BSA and PF127, and then subjected the samples to rheological analysis on round droplets. Our findings demonstrate that, for our system, the impact of passivation on condensate rheology is small compared to the TT dimerization-induced viscoelastic changes that we are investigating.

Figure 1. Rheological analysis of PT40 samples on BSA and PF127 passivated surfaces.

3. It's important to provide the power density of the UV illumination.

Response:

Thanks for this comment. In response to this comment, we measured the power density and provided the results in the revised manuscript.

4. The extensive use of condensate morphology (eg, circularity, semi-fusion, elongated shape) is concerning. This is because A), the morphology of condensates are strongly affected by the adhesion between condensates and the substrate (this adhesion is clearly not removed for most experiments). For example, see (<https://www.freeimages.com/premium/water-droplets-on-a-metal-surface-1592483>). B) condensates with similar circularity can have very

different mechanical properties, as illustrated in several recent studies (for example, <https://pubmed.ncbi.nlm.nih.gov/39211102/>).

Response:

Thanks for this comment. While we agree that surface properties impact the morphology of condensate droplets, our comparative analysis is focused on UV-induced changes under constant surface chemistry. As such, the morphological changes can be attributed to DNA-induced thymine dimerization within the condensates. To better demonstrate this, we performed new experiments where we generated condensates on two different chemistries, with high and low adhesiveness, and subjected the condensates to UV illumination. The UV-induced morphological changes are apparent and can be distinguished from surface effects. More specifically, we performed the UV irradiation experiment on surfaces passivated with Pluronic F-127 (PF127), a coating known to produce non-sticky droplets (as shown in the current work and consistent with Figure 6A) (See Yao, et al., PNAS 121(22), (2024)). Following 45 minutes of UV irradiation on these PF127-passivated dishes, we observed the formation of fusion-arrested droplets with significant morphological changes, as we observed on the BSA-passivated surfaces. This evidence supports the conclusion that the morphological changes and the formation of doublets are directly attributable to the UV-induced thymine dimerization within the condensates and the associated changes in material properties. We have included these data in a new Supplementary Figure S2, which shows consistency with Figure 2B.

Finally, we do agree with the referee that making a causative link between morphology and mechanics is not straightforward in general. We rephrased the corresponding parts in our manuscript to avoid confusion. In our study, however, we do see morphological and mechanical changes upon UV illumination, and the crosslinking of polymers has also been previously reported to affect mechanics and morphology in other condensate chemistries (See e.g., Doi: 10.1016/j.xcrp.2025.102430)

5. The authors should better explain the size distribution of condensates and their corresponding formation mechanisms. These are unlikely common knowledge for the readers. Along this line, how does condensate surface tension affect its size distribution?

Response:

Thanks for this comment. We do agree that we have been too short in our description, which was due to the journal's article length limit. In the revised manuscript, we elaborated on size distribution analysis and formation mechanics. Given the space limitation, we added this largely to the description of Figure S2.

To address the comment on surface tension, we measured surface tension using SPM for both +UV45 and PT40 droplets and included them in the revised article.

UV-induced thymine dimerization alters both the surface tension and the rheology of the condensates, leading to dynamic changes in droplet size over time. An increase in surface tension promotes coalescence, contributing to the growth of large droplets, while intermediate-size droplets decrease as they merge into larger structures. By 45 minutes, very large, fusion-arrested droplets appear, reflecting rheology-mediated arrest of coalescence, while the smallest droplets continue to form via fragmentation during incomplete coalescence events. Moreover, because illumination begins earlier in the +UV45 assay than in +UV30, a larger population of small pre-existing droplets becomes crosslinked and preserved in +UV45, contributing to the increase observed in the smallest size bin relative to +UV30. This overall pattern is clearly reflected in the size distributions presented in the manuscript.

6. In figure 2, the observation that the crossing point between G' and G'' increase between 30 and 45 min UV is surprising. How does this reconcile with the solidification of the condensate and the observed change in terminal viscosity (which itself needs to be better defined)?

Response:

Thanks for this comment. Upon moderate UV exposure, G' and G'' increase dramatically and a crossover appears, indicating a transition from liquid-like to solid-like behavior. At higher UV doses, both G' and G'' continue to increase, but the crossover shifts to higher frequencies. This shift arises from the formation of a heterogeneous network with locally crosslinked domains, which increases short-time stiffness and allows faster relaxation within these smaller domains. The condensate becomes stiffer overall, yet the network relaxes faster at the timescales probed, because motion is confined to these local regions. Unlike a fully percolated network characteristic of a complete liquid-to-solid transition, this state represents a partial liquid-to-solid transition, or incipient gelation. The shift of the crossover to higher frequency reflects a shorter dominant relaxation time, which falls outside the measurement window. We added an explanation about this in the revised article.

We revised the part on terminal viscosity to ensure that its definition and the procedure behind its derivation is clearly stated.

7. Does aging play a role in the liquid to solid transition of the PT40 condensates? For example, it would be better to include a set of data a separate plot to show the material properties of PT40 is stable for at least 45 min.

Response:

Thanks for this comment. To address this question, we performed new experiments which reveal that the aging effect is negligible within the aforementioned time scales, as can be seen from the figure below. We added a phrase to the revised paper on this point.

Figure 2. Rheological analysis of PT40 samples 45 minutes aged after 4 hours of standard incubation time.

8. A major point of the paper is thymine dimerization induced by UV. While plausible, we didn't find any direct evidence in the paper to support this claim. Could the authors check with electrophoresis?

Response:

Thanks for this comment. To address this question, we performed the requested electrophoretic mobility check, as well as two other complementary studies. To perform this study, we released the DNA from the condensates by introducing salt and screening the charged interactions. The droplets got dissolved, and the solution was then subjected to MST (See New Figure 4I). The results clearly show that +UV45 DNAs are diffusing significantly slower than the control, indicating a larger molecular size and crosslinking. UV treatment in Premix condition led to exclusively intrachain TT bonds, which leads to reduced gyration radii and higher mobility. As another way of proving TT dimer formation, we performed UV spectroscopy, which clearly shows formation of TT bonds in agreement with previous reports (New Figure 1H). Finally, we set up an anti-TT dimer antibody assay, which showed clearly staining of UV-treated droplets and lack of staining of control ones (New Figure 1I). These new studies provide direct evidence for the formation of the TT dimer in our samples. We included these new results in the revised article.

9. The observation of compartments within condensates is interesting. But this result is very descriptive and lack of mechanistic insights.

Response:

Thanks for this comment. The compartmentalization phenomenon we observe is identical to that reported by Erkamp et al. (Ref 56). In that work, it was determined

across numerous systems that internal cavities formed when the environmental conditions were changed such that the condensate favored a more compact/dense state. If the condensate contraction occurred faster than the molecules could redistribute, the rapid contraction would create cavities of dilute phase within the dense droplets. The study of Erkamp et al. used temperature drops and salt additions to reduce the repulsion between like-charged components (reducing the temperature causes a decrease in the Donnan pressure of the neutralizing counterions). Given the strong similarities between our system and Erkamp's, we are confident that we are observing the same phenomenon. There are two mechanistic details, however, that differ in our system. First, the removal of salt in our system promotes the association of oppositely charged components. Second, the presence of UV-induced crosslinks will lead to much longer redistribution timescales compared to the reptation-limited timescales of Erkamp et al. We have extended the discussion of this phenomenon in the manuscript to strengthen the connection with reference 56.

10. The fusion between two condensates is affected by condensate viscosity. So, it's unclear if data in figure 5 can be used to draw direct conclusion on condensate surface tension and if it is affected by the speed of the measurement. Plus, the claim of 'bridge formation' does not appear to be directly supported by imaging evidence.

Response:

Thanks for this comment. Extend speed does affect the maximum adhesive force (as shown in the new data provided in Figure S7. In our study, we operate under constant speed, and the changes in fusion dynamics are due to surface effects, geometry, and rheological properties. To investigate these effects and their contributions, we performed new analyses and developed new models, which are presented in the revised article (See Section on SPM analysis of droplet fusion dynamics, as well as Supplementary Notes S2 and S3). We measured surface tension using an SPM-based approach as well as using the capillary model for the case of control droplets. In +UV45 droplets, the relation between maximum adhesive force and surface tension deviates from the capillary models and can be explained by the models that are adjusted for viscoelastic effects.

Regarding bridge formation: our setup does not have side imaging, and thus, it is not possible to provide imaging evidence. As such, we revised our text to acknowledge this point.

Minor problems:

1. Line 82 'preserving chemical composition' is inaccurate since dimerization is a chemical change.

Response:

Thanks for this comment. Of course, we agree that addition or rearrangement of chemical bonds is indeed a chemical change. Our intention in using the phrase “preserving chemical composition” was to distinguish our approach from more common strategies in which condensates are chemically modified through the introduction of new molecules or crosslinkers, leading to changes in composition. In our study, no additional molecules or crosslinkers are introduced into the condensate, thus “atomic (or elemental) composition” is preserved, and chemical composition changes are limited to covalent bond formations (connectivity). We have revised the text to clarify this distinction while still conveying our intended point more accurately.

Revised introductory phrase reads:

“... while preserving chemical composition —that is, without adding new molecules and without changing atomic composition.”

Revised phrase in the discussion reads:

“... light can be leveraged to program the connectivity and internal organization of a nucleic acid-based condensate system, thereby changing condensate mechanics with no alterations to atomic (elemental) composition.”

2. In line 167, the author mentioned “this condition produced droplets of convenient size for mechanical analysis” but did not state what is the actual size in the main text or main figures.

Response:

Thanks for this comment. We revised this phrase, and the size information is included in the manuscript (methods section) and will be deposited in the associated raw data as well.

3. Figure 1A and many other images are missing scale bar.

Response:

Thanks for this comment. We added the missing scale bars.

4. Figure 1F. I would recommend defining what is single/paired/cluster with images.

Response:

Thanks for this comment. This is now included in the revised version, in Figure 2A. Please note that in the revision, we refrained from using two interchangeable terms, doublet and paired. All such structures are now referred to as doublets.

5. Line 232, unclear if the values were quoted for a specific frequency.

Response:

Thanks for this comment. We fixed this issue in the revised version.

6. Line 246, reference missing.

Response:

Thanks for this comment. We added a reference.

7. The description and alignment of Figure 3A and 3B are overall confusing. Since A and B are two different experiments. It is better to make separate text descriptions for them under figure 3.

Response:

Thanks for this comment. We fixed this issue in the revised version, and fully separated the two descriptions and figure panel arrangements. Please see the new Figure 4.

8. Although “applying UV immediately after mixing (+UV45_0h)” was observed with no condensate formation in 3A, it still recommended to include a confocal image of +UV45_0h with 1% TYE-665 tagged dT40 in Figure 3B.

Response:

Thanks for this comment. We performed new confocal imaging as well as MFI analysis, which are included in Figure S4.

9. Figure 3F meant to show no significant differences between control and premix samples but the control is not plotted.

Response:

Thanks for this comment. We included the control data in the plot, showing that the differences are relatively minimal.

10. Line 394, the use of reduced radius is unclear.

Response:

Thanks for this comment. We checked this part and ensured that the definitions and approach are clearly stated.

Reviewer #2 (Remarks to the Author):

Response:

Thank you for your time and constructive feedback, which helped us improve the manuscript. We have extensively revised the article, incorporating new experimental data, theoretical models, and additional mechanistic insights that not only strengthen the study but also highlight its broad significance and potential impact across chemistry,

materials science, and biomolecular condensate research, consistent with the scope and readership of Nature Communications.

Reviewer #3 (Remarks to the Author):

In the submitted manuscript, scanning probe microscopy is combined with optical microscopy to study the mechanical properties and fusion dynamics of condensate droplets. The droplets are formed in solutions of two polyelectrolytes, single-stranded DNA consisting of 40 thymine nucleotides (oligo-ssDNA dT40), which carries 40 negative charges, and the protein PLL with an average length of 150 residues, carrying 150 positive charges. The solution undergoes phase separation within a certain concentration regime as characterized in Figure S1.

When the polyelectrolyte solution is illuminated with UV light, the two polymers form crosslinks which can drive the droplets from a liquid to a gel-like or solid-like state, thereby changing the mechanical properties of the droplets and their morphology. The transition from the liquid to a gel- or solid-like state can be used, for instance, to arrest the fusion of the condensate droplets in some intermediate droplet state with nonspherical morphology as shown in Figure 1G.

My main concern is that the results reported in this manuscript are quite reasonable from an intuitive and qualitative point of view whereas the analysis of these results is rather fragmented, often confusing, and difficult to follow.

I recommend major revision of the submitted manuscript. This revision should, in particular, address the following points, ordered in the same way as the corresponding manuscript pieces in the main text and in the supplementary information:

Response:

Thanks a lot for the critical assessment of the paper and positive remarks on the findings. We carefully studied the comments and did our best to address them all. The paper is also restructured and rewritten in numerous parts to improve readability and flow. In what follows, we describe the revisions we did to the manuscript:

1) The title of the manuscript emphasizes thymine dimerization and the abstract claims on page 1, line 27 and 28, that "UV-induced thymine dimerization alters condensate nucleation and coalescence". However, this claim remains obscure until Figure 6 at the end of the manuscript. Therefore, I urge the authors to mention the competition between covalent intra- and interchain contacts, which is currently

explained in Figure 6, already in the abstract and to further clarify the text pieces before and after Figure 6.

Response:

Thank you for this comment. We have revised the abstract to explicitly mention the competition between intra- and inter-chain contacts. In addition, we have reworked the text associated with Figure 6 (now Figure 7) and the sections before it. The concept of balance between inter- and intra-chain contacts now appears consistently throughout the paper. We also include substantial new experimental data and theoretical results to strengthen this point. In particular, we added a new Figure S11, which complements the model presented in Figure 7.

2) At the end of the abstract, the authors claim that their findings have "implications for genome biology, the emergence of life on Earth, and engineering applications." I can see possible engineering applications, related, for instance, to synthetic biology but I have severe problems to imagine implications for genome biology and for the emergence of life on Earth. In fact, "genome biology" is not mentioned anywhere else in the manuscript and the "origin of life" is only briefly mentioned in line 446 on page 17 at the very end of the manuscript. If the authors want to maintain these claims, they should tell the interested reader in much more detail what they have in mind.

Response:

Thank you for this comment. We agree that our original descriptions were too brief and somewhat overstated. Our study is primarily focused on synthetic biology and materials science, with some discussion of possible implications for the origin of life. Indeed, UV-induced dimerization is a pathological process and therefore has no role in genome physiology; consequently, our reference to genome biology did not accurately reflect this point. In the revised manuscript, we have removed the reference to genome biology and revised the text throughout to elaborate on these broader points, while remaining within the journal's length limits.

3) Related to point 2): Condensate droplets in living cells have been discovered in 2009 and have been intensely studied during the last two decades by a variety of experimental and theoretical methods. It is now understood that intracellular condensate droplets are usually in a liquid-like rather than in a gel-like or solid-like state. Therefore, do the authors really believe that gel-like or solid-like condensate droplets play some important role in living cells? If yes, they should explicitly describe their line of reasoning in the manuscript.

Response:

Thank you for this comment. We agree that condensates identified in healthy human cells are typically liquid-like. Gel-like condensates are frequently observed in pathologies and aging processes. The origin-of-life aspect of our study considers primitive life forms, and there is evidence that condensates in bacteria can adopt gel-like states. We have elaborated on these points in the revised manuscript (and included supporting citations) to avoid potential confusion.

4) The first paragraph of "Materials and Methods" provides some information about the preparation of the stock solutions. This information is, however, not sufficient if someone wants to repeat this preparation.

As far as dT40 is concerned, one would like to know about the buffer used to rehydrate the dry dT40 obtained from the producer. In particular, which salt was added at what concentration? And was the pH controlled during this preparation?

As far as the PLL protein is concerned, I would have expected that this protein was produced by some recombinant method, which would lead to a well-defined length of the protein.

However, in line 101 and 102 on page 4, the authors mention the "average" molecular weight and the "average" length of the PLL protein. Where does the polydispersity of PPL come from and how large is it?

Response:

Thank you for this comment. We agree with the referee and apologize for the brevity in our original descriptions. In the revised manuscript, we have included practical details, such as the buffer composition, salt concentration, and pH control during the rehydration of dT40, to ensure reproducibility of the study.

We have also clarified our description of PLL. We intentionally chose to use PLL with a range of lengths as a simple way to model the diversity of positively charged polypeptides found in cells or in primitive life forms. Please note that the polymer exhibits a distribution of lengths due to its synthesis protocol. Fully defined-length PLL can be obtained through synthetic routes, but such approaches are typically limited to short chains and small quantities, not feasible for condensate studies. We clarified these points in the revised version of the manuscript (See Materials and Methods).

5) After cross-linking the two polymers by UV illumination, the droplets exhibit an increased viscosity as reported in Figure 2E. This increase is dramatically different depending on the protocol used for the illumination. The lettering at the x-axis of Figure 2E distinguishes three cases: "control", "30minUV:4hrs", and

"45minUV:4hrs".

One might think that "xyminUV:4hs" stands for "applying UV for xy min after 4 hours", but these abbreviations are not explained in the text. Instead, in line 241 on page 9, the same protocols are called "+UV30" and "+UV45". Furthermore, in line 249 on page 9, the authors say that the UV irradiation was applied "for 45 minutes" without mentioning the alternative exposure time of 30 minutes. In addition, the authors now refer to the "Premix" condition and to the "0 hours" condition.

Therefore, the description of the different protocols as used in the submitted manuscript is very confusing and must be improved. The same comment applies to Figure 3.

Response:

Thank you for this comment and for bringing these issues to our attention. We revised our abbreviation/terminology to avoid confusion.

6) Related to point 5): One general advantage of light-induced changes of molecules and chemical reactions is that these changes can be rather fast. Indeed, the publication by Johnson and Wiest, JPC B (2007), Ref 11 of the manuscript, emphasizes that "thymine dimerization is an extraordinarily rapid reaction." In contrast, the time scales discussed in the submitted manuscript for the UV illumination protocols are very long, corresponding to 30 or 45 minutes. Thus, the authors should explain why the UV-induced dimerization studied here is so much slower.

Response:

Thank you for this insightful comment. We agree that thymine dimerization is intrinsically very fast, as noted in Johnson and Wiest (J. Phys. Chem. B, 2007, Ref 25). For dimerization to occur, the bases must rotate into the correct orientation to form a cyclobutane dimer. For intermolecular dimerization, there is an additional limitation coming from the time required for the molecules to diffuse together. Such intermolecular reactions were studied by Nagpal et al. (Ref 49), who found that the reaction reached saturation on a timescale on the order of 20 minutes. These results are not directly applicable to our system for two reasons. First, the thymine monomers in the Nagpal experiments were at much lower concentrations than the interior of our condensates. Second, their system did not include phosphate backbones and, therefore, did not exhibit the strong electrostatic repulsion between nucleic acid chains. However, these two effects modify the dimerization rate in opposite directions; reduced concentration decreases the rate, and the lack of phosphates increases the rate relative to our system. Therefore, our empirical observation that UV continues to affect condensate properties past 30 minutes is consistent with the Nagpal results. We have included a brief discussion of these timescales in the manuscript.

7) The results for the apparent interfacial tension are displayed in Figure 5H in the form of a histogram. This histogram exhibits at least two, maybe even three peaks. The authors should explain whether or not they can interpret this histogram in terms of the underlying molecular processes.

Does this histogram reflect, for instance, the competition between covalent intra- and interchain contacts of the polymer chains?

Response:

Thank you for this comment. To address this question, we fully reworked this part of the manuscript and included new experimental results and developed new models that enable us to explain the observed heterogeneities. Figure 5H (now moved to Figure S10 B) is a histogram of $F/4\pi R^*$, which is the apparent interfacial tension for a liquid-like droplet (i.e., the PT40 samples). In +UV45 droplets, crosslinking induces large viscoelasticity changes and thus deviation from the capillary model. UV illumination is expected to alter surface tension too. Therefore, $F/4\pi R^*$ is a function of both viscoelasticity and surface tension together, and not surface tension alone. To demonstrate this directly, we measured surface tension for PT40 and +UV45 droplets and developed a modified version of the capillary model, which includes viscoelastic effects. Our analysis shows that UV treatment increases interfacial tension, and the spread of $F/4\pi R^*$ is due to heterogeneity of surface tension to a large extent, with some contribution from heterogeneity of the viscoelastic moduli. We further elaborate on this point in Note S3, where we also include a propagation analysis of these effects. Finally, we asked how the competition between covalent intra- and inter-chain contacts affects heterogeneity. To address this, we performed fusion experiments on Premix droplets and incorporated the results (along with the results of +UV45 droplets) into new theoretical models to estimate how the balance of inter- and intra-chain bonds influences noise propagation. Please see the last paragraph of the Results section.

8) Two typos on page 15: in line 399, Figure 5H should be replaced by Figure 5G, and in line 402, Figure 5I should be replaced by Figure 5H.

Response:

Thank you for this comment. We fixed these issues in the revised paper.

9) Figure S1 is quite useful in order to understand the formation of the biomolecular condensates studied here, Therefore, the authors might want to move Figure S1 into the main text.

Response:

Thank you for this comment. Following your advice, we included this figure in the main text, as new Figure 1A.

Reviewer #4 (Remarks to the Author):

In a nutshell, the paper reports that UV illumination can alter the fusion times of condensate droplets. I haven't studied the literature exhaustively, but this does seem to me like a new and interesting discovery. I have comments:

Response:

Thank you very much for the critical assessment of the paper and positive remarks on the findings. We carefully studied the comments and did our best to address them all. In what follows, we describe the revisions we did to the manuscript:

- I am not an expert in this specific area, but the authors state with great confidence that the observed changes in condensate mechanics are due to thymine dimerization. I do not see direct evidence for this claim; at present it seems more like a plausible hypothesis rather than a demonstrated molecular mechanism.

Response:

Thank you for this comment. To address this comment, we decided to take three experimental approaches to probe TT dimerization:

- 1- We developed an assay in which we used anti-TT dimer antibodies to directly probe TT dimer formation within the condensate phase (See Figure 1I).
- 2- We performed UV spectroscopy, which revealed the spectroscopic signature of TT dimer formation (See Figure 1H).
- 3- We developed a microscale thermophoretic (MST) assay, which reveals a dramatic change in DNA mobility upon UV treatment, which is expected for cross-linked DNA molecules (See Figure 4I).

These three lines of evidence prove the formation of the TT dimer in our study. The new results are included in the revised version.

- In Fig. 1G and H the authors analyze “neck length.” While this is a reasonable empirical observation, is there a quantitative framework that could connect this parameter to rheological or other mechanical quantities?

Response:

Thank you for this comment. In classical Newtonian fluids, neck growth during early-stage coalescence scales with the ratio of surface tension γ to viscosity η as $r(t) \propto (\gamma/\eta)t$ in the viscous-dominated regime, where t is the time since contact; this dynamics is not strongly influenced by elasticity [See P. J. Dekker et

al., Phys. Rev. Lett. 128, 028004 (2022)], and so looking at the neck growth in time with a higher spatial and temporal resolution may provide information about the viscosity. However, highly time- and space-resolved analysis is challenging in condensate assays, and the dense viscoelastic environment of the condensates complicates interpretation.

In our study, the previously named “neck length” corresponds to a 2D projection of the contact region, which relates to the 3D neck radius in a complex way (and differs from the early-stage neck radius used in scaling relations). To avoid confusion, we have renamed this parameter as “chord length” in the revised manuscript. We focus on relative comparisons of chord length under different conditions rather than extracting absolute rheological parameters from it. Extended UV illumination increases chord length, reflecting slower relaxation due to higher viscoelasticity, which also allows doublets with larger chord lengths to be more readily captured during imaging. Rheological properties are instead quantified directly via SPM measurements. We have clarified these points in the revised manuscript to contextualize chord length as an empirical, indirect indicator of mechanical state.

- I find the introduction poorly written and the results written with a somewhat overstated style. For example:
 - o Micropipette aspiration does not, in general, require the use of tracer particles.

Response:

Thank you for this comment. We agree that this part of the intro was not fully clear and in some parts overstated. We extensively revised the introduction and the discussion sections to improve clarity and tone.

Regarding micropipette aspiration, you are correct that it does not require tracer particles. The original statement was an error introduced during writing and does not reflect our understanding of the technique. Our intention was to highlight its limitations in force range and experimental throughput, and the revised manuscript now states this accurately.

o To my knowledge, thymine dimerization involves the formation of covalent bonds, which necessarily alters the chemical composition. Therefore, the sentence claiming that chemical composition remains unchanged should be revised.

Response:

Thanks for this comment. We agree that addition or rearrangement of chemical bonds is indeed a chemical change. Our intention in using the phrase “preserving chemical composition” was to distinguish our approach from more common strategies in which condensates are chemically modified through the introduction of new molecules or crosslinkers, which alter elemental composition. In our study, no additional molecules or crosslinkers are introduced into the condensate, and salt concentrations remain unchanged. Thus, while

covalent connectivity is modified, the elemental (atomic) composition of the condensates remains unchanged. We have revised the text to clarify this distinction while still conveying our intended point more accurately.

Revised introductory phrase reads:

“... while preserving chemical composition —that is, without adding new molecules and without changing atomic composition.”

Revised phrase in the discussion reads:

“... light can be leveraged to program the connectivity and internal organization of a nucleic acid-based condensate system, thereby changing condensate mechanics with no alterations to atomic (elemental) composition.”

o The text contains multiple “first-of-its-kind” statements that are unnecessary.

Response: Thank you for this comment. We fixed this issue in the revised version and removed all such references.

o Prior analogous work should be acknowledged, for instance the use of AFM to measure the material properties of objects as large as condensates (<https://www.nature.com/articles/nphys4104>).

Response: Thank you for this comment. To address this issue, we added the suggested reference and a few others, including a recent technical paper that covered prior work extensively.

o The paper is about material properties and their relation to molecular conformations, but does not cite the significant very recent progress in this area from the Banerjee, Mittal, and Schuler groups.

Response: Thank you for this comment and for bringing this interesting study to our attention. We agree that significant recent progress by the Banerjee, Mittal, and Schuler groups is relevant to our study. We have now included citations to these works in the revised manuscript.

• Line 332 seems to be referring to Figure 5B. Please check.

Response: Thank you for this comment. We addressed it in the revised version.

• Finally, I find it puzzling that no condensates are reported in the “0h” condition. Could it be that phase separation does occur, but the droplets are arrested at sizes below the diffraction limit of the imaging system?

Response: We thank the reviewer for this insightful comment. Indeed, we cannot completely rule out the presence of condensates smaller than the diffraction limit during

the early phases of the “0 h” condition. Capturing the very first moments under UV illumination is practically challenging, as small objects typically remain floating and require time to settle onto the substrate before they can be imaged.

To address this, we took two approaches (See Figure S4):

- 1) Real-time phase contrast imaging of objects on the substrate, during UV illumination, from time = 0 to time = 45 min.
- 2) Microfluidic Imaging (MFI) analysis, which allows measurement of particle size and circularity while particles are freely floating. This was performed after the UV illumination was completed.

Our phase contrast imaging revealed the formation of tiny condensate-like droplets at early stages, which then quickly transitioned to aggregate-like structures within the time frame of the experiment. MFI analysis revealed particles with heterogeneous sizes and circularities. A subset of these particles appeared somewhat circular, but notably less circular than all condensates observed in our study under other conditions.

Finally, we note that TT dimer formation in the dilute phase occurs on a very short timescale (See our note in the Results section), which implies that condensation formation may be affected almost immediately upon UV exposure. As such, while the condensate-to-aggregate transition is a process we observe, direct nucleation and growth of the aggregate states may also occur. However, capturing and characterizing small and transient structures is technically challenging. We have revised the manuscript to acknowledge this possibility explicitly and to highlight the limitations imposed by the diffraction limit of optical microscopy, and incorporated our new experimental data.

REVIEWERS' COMMENTS

Reviewer #1 (Remarks to the Author):

The authors have largely addressed our concerns in the revised manuscript.

Response: We thank the referee for the constructive feedback on our manuscript and are glad that the revisions were found satisfactory.

We only have the following suggestions:

1. Figure 1 in the response letter show a ~5 fold difference between two coating conditions. The data do not exclude the possibility the coating can have a significant effect the measured viscoelasticity of condensates. We think the authors should be more cautious about this point and include this result as a supporting information in the manuscript.

Response: Thanks. We agree that surface properties does influence mechanical properties as seen in our data. As suggested by the referee, we included the rheology data on PF127-coated surface (presented in our previous response letter) as a new panel to Figure S2.

2. We were unable to locate the power density values of the UV illumination.

Response: The power density is given in the second paragraph of Results section. It reads as follows:

*For UV treatment, samples were prepared over a total period of 4 hours and placed under a UVC lamp (UVP UVG-11, $\lambda=254$ nm; **1120 μ W/cm² at the sample**) (Figure 1D), with UV illumination applied during the final 30 (+UV30) or 45 (+UV45) minutes⁴⁹.*

3. The authors claim that SPM has higher range of forces compare to OT and micropipette aspiration without reporting the force range of SPM. But this is not supported by any direct evidence.

Response: Thanks. We included the force range available through SPM in the Methods section.

4. The number of samples and independent measurements are missing in several figures, including figure 2 and s2.

Response: We thank the reviewer for this comment. We have thoroughly reviewed the manuscript and incorporated the relevant sample size information where appropriate.

Reviewer #2 (Remarks to the Author):

I co-reviewed this manuscript with one of the reviewers who provided the listed reports. This is part of the Nature Communications initiative to facilitate training in peer review

and to provide appropriate recognition for Early Career Researchers who co-review manuscripts.

Response: We thank the referee for the constructive feedback on our manuscript.

Reviewer #3 (Remarks to the Author):

The authors have responded to all concerns raised in my previous report and made appropriate changes in the revised manuscript. The manuscript is now suitable for publication in Nature Communications.

Response: We thank the referee for the constructive feedback on our manuscript. We are pleased that the referee is satisfied with our revisions and has recommended publication of the work.

Reviewer #4 (Remarks to the Author):

Good for publication.

Response: We thank the referee for the constructive feedback on our manuscript. We are pleased that the referee is satisfied with our revisions and has recommended publication of the work.